# Learning Recourse on Instance Environment to Enhance Prediction Accuracy

**Lokesh Nagalapatti** *    **Guntakanti Sai Koushik**    **Abir De**    **Sunita Sarawagi**
Department of Computer Science and Engineering
IIT Bombay

## Abstract

Machine Learning models are often susceptible to poor performance on instances sampled from bad environments. For example, an image classifier could provide low accuracy on images captured under low lighting conditions. In high stake ML applications, such as AI-driven medical diagnostics, a better option could be to provide recourse in the form of alternative environment settings in which to recapture the instance for more reliable diagnostics. In this paper, we propose a model called RECOURSENET that learns to apply recourse on the space of environments so that the recoursed instances are amenable to better predictions by the classifier. Learning to output optimal recourse is challenging because we do not assume access to the underlying physical process that generates the recoursed instances. Also, the optimal setting could be instance-dependent — for example the best camera angle for object recognition could be a function of the object's shape. We propose a novel three-level training method that (a) Learns a classifier that is optimized for high performance under recourse, (b) Learns a recourse predictor when the training data may contain only limited instances under good environment settings, and (c) Triggers recourse selectively only when recourse is likely to improve classifier confidence. We experiment with synthetic and real world datasets to show the efficacy of our proposed approach.

## 1   Introduction

The performance of any supervised learning model depends strongly on the quality of input instances. However, in practice, instances may be of suboptimal quality when generated in adverse environment settings. For example, even an expressive image classification model may misclassify an image shot at an extreme close-up or at a wrong angle or under poor lighting [13, 28]. Despite large training sizes, such unfavorable instances can deteriorate model performance which can have serious consequences in high stake scenarios like AI guided crop monitoring [21], automatic disease diagnosis from images [22], and AI driven accessibility enhancement for the hearing impaired.

Mitigating the effect of such unfavorable instances entails the design of recourse mechanism to recommend alternative environment *settings* that yield instances revealing the target class. For example, in low cost smartphone based medical diagnosis [22] where imaging is performed by non-experts, such recourse mechanisms can interactively recommend camera settings that yield images optimal for the downstream diagnosis model. Recourse could be particularly useful for healthcare on the edge where users can be prompted to adjust their edge-device settings in real-time to deploy the diagnosis model with higher accuracy. The optimal camera settings however could be label dependent. For example, the best camera angle for recognizing an aeroplane could be different from the angle for recognizing poles.

More formally, the problem that this paper seeks to address is as follows. We have an object $z$ in the physical space (e.g. a crop) with an unknown true label $y$ (e.g. type of disease). Let $\beta \in \mathcal{B}$ be

---

* nlokeshiisc@gmail.com

36th Conference on Neural Information Processing Systems (NeurIPS 2022).

the environment setting under which we capture a digital representation $\mathbf{x}$ of $z$ to diagnose the label from a downstream classifier $f_\theta(\mathbf{x})$. Our goal during recourse is to recommend an alternative setting $\boldsymbol{\beta}'$ (if any) to the user for getting a different representation $\mathbf{x}'$ of $z$ where $f_\theta(\mathbf{x}')$ is more likely to be correct than $f_\theta(\mathbf{x})$. The above problem is an instance of algorithmic recourse, on which there has been much recent work [30, 29, 23, 8, 32, 11]. These methods recommend recourse actions on the instance space $\mathbf{x}$, which is difficult to realize on raw data for objects such as images and speech. Instead we propose to intervene at the level of the environment which generates the instance via an unknown physical process. We view our contribution under three facets as explained below:

(i) **Novel framework for recourse mechanism.** We propose RECOURSENET, a trainable recourse mechanism which recommends modified actions to the end user so that, if acted upon the environment, it can generate instances with improved accuracy. RECOURSENET consists of three components: (1) a classifier $f_\theta$, (2) a recourse trigger $\pi$ (3) a recourse recommender network $g_\phi$. Given an instance $(\mathbf{x}, \boldsymbol{\beta})$, the recourse trigger $\pi$ first decides whether to recommend recourse for $\mathbf{x}$. If so, the recourse recommender $g_\phi$ suggests an alternative environment $\boldsymbol{\beta}'$. Using these, the user generates a new instance $\mathbf{x}'$, on which $f_\theta$ would give the correct label with potentially higher confidence.

(ii) **Three level training proposal.** The main challenge of RECOURSENET is that we do not assume access to the latent physical process $Z$ that generates an $\mathbf{x}'$ given a $\boldsymbol{\beta}'$ during training. Instead we train with a fixed labeled dataset containing (latent) objects $z_i$ rendered as instances $\{\mathbf{x}_{ij}\}$ under a small but variable set $B_i$ of observed settings $\{\boldsymbol{\beta}_{ij}\}$. We show that direct end-to-end training of a combined likelihood training settles on easy local minima, and fails to provide good recourse. Training them stage wise also is challenging; we list some of these. For $f_\theta$, training on the entire dataset may be suboptimal since instances in poor settings, where recourse will be asked, may mislead decisions on good instances. For $g_\phi$, we have no direct supervision of good $\boldsymbol{\beta}$ for a given $(\mathbf{x}_{ij}, \boldsymbol{\beta}_{ij})$. For $\pi$, simple heuristics like choosing to recourse examples where $f_\theta$ has low confidence does not guarantee improved accuracy. Our training strategy employs careful scheduling and decoupling of the training of the three modules via proxy functions. This achieves substantial gains over simple end-to-end training and existing methods of training classifiers with data selection based purely on noise [20, 2, 5, 12, 14, 17, 24, 31].

(iii) **Characterization of recourse conditions.** We provide theoretical characterizations to identify the circumstances under which recourse will enhance prediction accuracy. Specifically, we show that given an instance $\mathbf{x}$, if the recourse recommender suggests a modified environment that is close to at least one of the training environments resulting in an improved accuracy, then the recourse is beneficial. Moreover, if there exists some environment which improves the accuracy by a substantial margin, then even a modestly calibrated recourse recommender can lead to improved accuracy.

## 2    Related work

Our work is closely related to (i) Algorithmic recourse, (ii) Learning with triage and (iii) Machine learning with environment perturbation.

**Algorithmic recourse**: In recent years, there is an increasing interest in designing recourse on the instance space [30, 29, 23, 8, 32, 11, 25, 9] for a wide variety of applications. For example [30, 29] aim to improve fairness; [15, 7, 10] aim to train the models so that the predicted output is preserved under strategic perturbation of the instance space. Another line of work called strategic classification [7, 15, 10] deals with applying causal interventions to instances. However, these work learn the recourse action on the instance space, whereas, our goal is to design recourse action on the observed environment. An additional challenge in our setting is that the impact of the environment on the instance is latent and we do not assume presence of enough labeled data to learn a generative model for complex real-world instances under different environments.

**Learning with triage:** A recent line of work [20, 2, 5, 12, 14, 17, 24, 31] aims to learn when to outsource a subset of instances to human and assign the rest of the examples to machine so that machine and human together achieve superior performance than what they would have achieved independently. However, in our problem, humans do not participate in prediction task but they only generate new instances under the recommended environments.

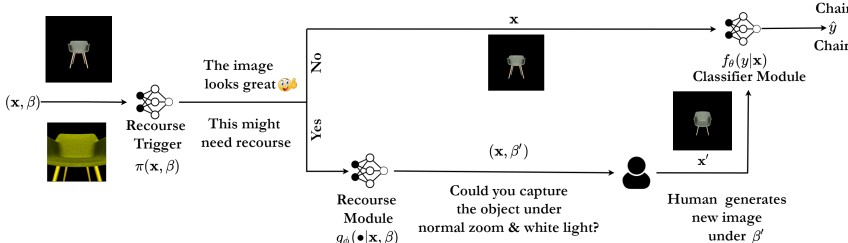

Figure 1: Architecture of Proposed Approach. The chair image on the top does not need recourse but the bottom image obtains the correct label only after recourse.

**Machine learning under environment perturbations:** Machine learning models are sensitive to environments under which data is generated [13, 28]. For example [13], show that simple parametric perturbations on the Shapenet dataset can flip class labels. In another related work [18] suggests interventions on the environment using policy gradients to train a recourse model. However, they assume the availability of a human through out the training loop to generate data in an on-demand basis. We make no such assumptions and train with a fixed labeled dataset.

# 3 Proposed approach

In this section, we first formally present our problem, present our training methodology, and then theoretically characterize the settings under which recourse is possible.

**Problem formulation** Let $\mathcal{Z}$ denote a space of objects, $\mathcal{B}$ denote a space of environment settings which could be real-valued or discrete or mixed, and $\mathcal{X}$ denote a space of instances obtained via a latent physical process $Z : \mathcal{Z} \times \mathcal{B} \to \mathcal{X}$. Given a latent object $z \in \mathcal{Z}$ and an environment setting $\boldsymbol{\beta} \in \mathcal{B}$, we get an instance $\mathbf{x} \in \mathcal{X} = \mathbb{R}^{d_x}$ *i.e.*, $\mathbf{x} = Z(z, \boldsymbol{\beta})$. Each object $z$ has a label $y \in \mathcal{Y}$ with $|\mathcal{Y}| = K$. We are interested in inferring the object's label using a trained classifier $f_\theta$. During training, for each of the latent set of objects $\{z_i\}_{i \in D}$, we are given a true label $y_i$ and for a small set of settings $B_i \subset \mathcal{B}$, we are given instance $\{\mathbf{x}_{ij}\}_{j \in B_i}$. Thus, we view the training data as a set of examples $T = \{y_i, \{\mathbf{x}_{ij}, \boldsymbol{\beta}_{ij}\}_{j \in B_i}\}_{i \in D}$ We use $V$ to index all the examples, *i.e.*, $V = \cup_{i \in D}\{\{i\} \times B_i\}$. As stated earlier, our goal is to design a recourse mechanism that given a representation $\mathbf{x}$ obtained of a latent object $z$ under given settings $\boldsymbol{\beta}$ will recommend an alternative $\boldsymbol{\beta}'$ if the resultant $\mathbf{x}' = Z(z, \boldsymbol{\beta}')$ is expected to yield more accurate prediction under $f_\theta$. Note that $Z$ is not accessible to us during training and we assume in this work that it is difficult to learn $Z$ or infer $z$ from the available labeled data $T$. Our goal instead is to use $T$ to learn both $f_\theta$ and the recourse mechanism.

## 3.1 Training RECOURSENET

RECOURSENET consists of three components:

1. A classifier $f_\theta : \mathcal{X} \times \mathcal{Y} \to [0, 1]$ which aims to capture the likelihood of the label $y$ given an instance $\mathbf{x}$, *i.e.* $f_\theta(y \mid \mathbf{x})$ approximates $\Pr(y \mid \mathbf{x})$.

2. A recourse recommender network $g_\phi : \mathcal{X} \times \mathcal{B} \times \mathcal{B} \to [0, 1]$, that suggests a modified environment $\boldsymbol{\beta}' \sim g_\phi(\bullet \mid \mathbf{x}, \boldsymbol{\beta})$ such that if the user (via $Z$) were to regenerate a new instance $\mathbf{x}'$ using $\boldsymbol{\beta}'$ the classifier is likely to provide higher accuracy.

3. A recourse trigger network $\pi : \mathcal{X} \times \mathcal{B} \to \{0, 1\}$ which is a binary decision function. Here, $\pi(\mathbf{x}, \boldsymbol{\beta}) = 1$ indicates that we decide to perform recourse on the environment and the $\boldsymbol{\beta}'$ suggested by $g_\phi$ should be used to regenerate the instance.

**Training objective.** Given a set of examples with $\{y_i, \{\mathbf{x}_{ij}, \boldsymbol{\beta}_{ij}\}_{j \in B_i}\}_{i \in D}$, we aim to find $\theta, \phi$ and $\pi$ by solving the following optimization problem:

$$\max_{\theta, \phi, \pi} \sum_{\substack{i \in D \\ j \in B}} \Bigg[ (1 - \pi(\mathbf{x}_{ij}, \boldsymbol{\beta}_{ij})) \log f_\theta(y_i \mid \mathbf{x}_{ij})$$
$$+ \pi(\mathbf{x}_{ij}, \boldsymbol{\beta}_{ij}) \log f_\theta(y_i \mid Z(z_i, \mathrm{argmax}_{\boldsymbol{\beta}} \, g_\phi(\boldsymbol{\beta} \mid \mathbf{x}_{ij}, \boldsymbol{\beta}_{ij}))) \Bigg] \tag{1}$$

$$\text{subject to,} \sum_{i \in D, j \in B} \pi(\mathbf{x}_{ij}, \boldsymbol{\beta}_{ij}) \leq b, \text{ and } \pi(\mathbf{x}_{ij}, \boldsymbol{\beta}_{ij}) \in \{0, 1\} \tag{2}$$

| **Algorithm 1:** GREEDYALGORITHM for training $f_\theta$ | **Algorithm 2:** TrainRECOURSENET |
|---|---|
| **Require:** Data $T = \{y_i, \{\mathbf{x}_{ij}, \boldsymbol{\beta}_{ij}\}_{j \in B_i}\}, b$ | **Require:** Train data $T = \{y_i, \{\mathbf{x}_{ij}, \boldsymbol{\beta}_{ij}\}_{j \in B_i}\}, b, \delta$ |
| 1: $V = \cup_{i \in D}\{\{i\} \times B_i\}$ | 1: $\hat{\theta} \leftarrow$ GREEDYALGORITHM$(T, b)$ |
| 2: $R \leftarrow \emptyset, \theta^0(\emptyset) \leftarrow$ TRAIN$(F(\bullet, \emptyset))$ | 2: $\hat{\phi}, f^{\mathrm{CF}} \leftarrow$ RECRECOMMENDER$(T, \delta, \hat{\theta})$  // Eq (7) |
| 3: **for** $k \in [b]$ **do** | 3: **Return** $\hat{\theta}, \hat{\phi}, f^{\mathrm{CF}}$ |
| 4:    **for** $(i, j) \in V \backslash R$ **do** | |
| 5:       $\mathcal{L}[(i,j)] =$ | **Algorithm 3:** RECOURSENET Inference |
|        $F(\theta^k(R \cup \{(i,j)\}), R \cup \{(i,j)\})$ | **Require:** Test instance $(z, \mathbf{x}, \boldsymbol{\beta})$, $Z$ (human), $\hat{\theta}, \hat{\phi}, f^{\mathrm{CF}}$ |
| 6:    $(i^*, j^*) \leftarrow \mathrm{argmax}_{(i,j) \in V \backslash R} \mathcal{L}[(i,j)]$ | 1: $\hat{\pi} \leftarrow$ RECTRIGGER$(\mathbf{x}, \boldsymbol{\beta}, \hat{\theta}, \hat{\phi}, f^{\mathrm{CF}})$  // Eq (8) |
| 7:    $R \leftarrow R \cup \{(i^*, j^*)\}$ | 2: $\hat{y} \leftarrow \mathrm{argmax}_y \big[(1 - \hat{\pi}(\mathbf{x}, \boldsymbol{\beta}))f_{\hat{\theta}}(y|\mathbf{x}) +$ |
| 8:    $\theta^{k+1}(R) \leftarrow$ TRAIN$(F(\bullet, R))$ |    $\hat{\pi}(\mathbf{x}, \boldsymbol{\beta})f_{\hat{\theta}}(y|Z(z, \mathrm{argmax}_{\boldsymbol{\beta}'} g_{\hat{\phi}}(\boldsymbol{\beta}'|\mathbf{x}, \boldsymbol{\beta})))\big]$ |
| 9: **Return** $\theta^{k+1}(R)$ | 3: **Return** $\hat{y}$ |

Here, $b$ indicates the maximum number of examples which can undergo recourse. The first term in the objective (1) $(1 - \pi(\bullet, \bullet)) \log f_\theta(\bullet \,|\, \bullet)$ accounts for examples that do not need recourse and the second term $\pi(\bullet, \bullet) \log f_\theta(\bullet \,|\, \bullet)$ accounts for those that need recourse. End to end training of the optimization problem (1)—(2) is challenging since we do not have an analytical form of $Z$ and training such a process will be difficult. We propose to train the three components $f_\theta, g_\phi, \pi$ in a carefully designed three-stage process that we describe next.

**Training the classifier $f_\theta$.** Training $f_\theta$ on the entire training data may be sub-optimal because instances in poor settings would be subject to recourse, and the classifier should instead focus on instances after recourse as the above training objective suggests. For training $f_\theta$ first we eschew the involvement of $Z$ and $g_\phi$ from the training objective (1) by noting that $\pi(\mathbf{x}_{ij}, \boldsymbol{\beta}_{ij}) = 1$ only if $f_\theta(y_i \,|\, Z(z_i, \mathrm{argmax}_{\boldsymbol{\beta}} g_\phi(\boldsymbol{\beta} \,|\, \mathbf{x}_{ij}, \boldsymbol{\beta}_{ij}))) \geq f_\theta(y_i \,|\, \mathbf{x}_{ij})$. Therefore, we replace the term $Z(z_i, \mathrm{argmax}_{\boldsymbol{\beta}} g_\phi(\boldsymbol{\beta} \,|\, \mathbf{x}_{ij}, \boldsymbol{\beta}_{ij}))$ with some instance $(\mathbf{x}_{ir}, \boldsymbol{\beta}_{ir})$ for some $r \in B_i$ of the same object $z_i$ such that the predicted classification accuracy on $\mathbf{x}_{ir}$ is better than the original instance $\mathbf{x}_{ij}$ by a certain margin $\Delta$. Specifically, given $(\mathbf{x}_{ij}, \boldsymbol{\beta}_{ij})$, we first define $\mathrm{Rec}_\Delta(\theta, \mathbf{x}_{ij}, y_i)$ as the set of environments which would improve the log-likelihood of the gold label by at least a margin $\Delta$ *i.e.*,

$$\mathrm{Rec}_\Delta(\theta, \mathbf{x}_{ij}, y_i) = \{\boldsymbol{\beta}' \in B_i \,|\, \log f_\theta(y_i \,|\, Z(z_i, \boldsymbol{\beta}')) > \log f_\theta(y_i \,|\, \mathbf{x}_{ij}) + \Delta\} \quad (3)$$

and then we pose the following training problem to learn $\theta$.

$$\max_{\theta, \pi} \sum_{\substack{i \in D \\ j \in B_i}} \left[ (1 - \pi(\mathbf{x}_{ij}, \boldsymbol{\beta}_{ij})) \log f_\theta(y_i \,|\, \mathbf{x}_{ij}) + \pi(\mathbf{x}_{ij}, \boldsymbol{\beta}_{ij}) \max_{\boldsymbol{\beta}_{ir} \in \mathrm{Rec}_\Delta(\theta, \mathbf{x}_{ij}, y_i)} \log f_\theta(y_i \,|\, \mathbf{x}_{ir}) \right]$$

$$\text{subject to,} \sum_{i \in D, j \in B_i} \pi(\mathbf{x}_{ij}, \boldsymbol{\beta}_{ij}) \leq b, \text{ and } \pi(\mathbf{x}_{ij}, \boldsymbol{\beta}_{ij}) \in \{0, 1\}. \quad (4)$$

Since our budget is limited, one needs to spend it on only those instances which not only suffer from poor accuracy, but can also lead to new instances that promote $f_\theta$ to predict the correct label. The presence of a non-zero margin $\Delta$ ensures such a condition. In Section 3.2, we provide the conditions under which such a recourse set will exist.

Given $\pi(\mathbf{x}_{ij}, \boldsymbol{\beta}_{ij}) \in \{0, 1\}$, we first define the set $R = \{(i, j) \,|\, \pi(\mathbf{x}_{ij}, \boldsymbol{\beta}_{ij}) = 1\}$. Then, we can write the objective (4) as

$$\max_{\theta, R:|R| \leq b} F(\theta, R) = \sum_{(i,j) \notin R} \log f_\theta(y_i \,|\, \mathbf{x}_{ij}) + \sum_{(i,j) \in R} \max_{\boldsymbol{\beta}_{ir} \in \mathrm{Rec}_\Delta(\theta, \mathbf{x}_{ij}, y_i)} \log f_\theta(y_i \,|\, \mathbf{x}_{ir}) \quad (5)$$

which gives us the problem of subset selection in conjunction with parameter estimation. Note that the involvement of $R$ as an optimization variable renders the above problem challenging even if $\log f_\theta(y \,|\, \mathbf{x})$ is concave in $\theta$. Thus, we resort to a greedy algorithm [19, 5, 16, 33] to solve this optimization problem (summarized in Algorithm 1). It is an iterative routine, which picks up an instance $(\mathbf{x}_{ij}, \boldsymbol{\beta}_{ij}, y_i)$ at every iteration which will maximize the training objective. Given an update $R$ at step $k \leq b$, it chooses a candidate instance $(i, j)$ which maximizes $F(\theta^k(R \cup \{(i, j)\}), R \cup \{(i, j)\})$, where $\theta(S) = \max_\theta F(\theta, S)$. We would like to highlight that, by definition of the set $\mathrm{Rec}_\Delta$, inclusion of $(i, j)$ in $R$ either improves the log-likelihood or keeps it at the same value obtained in the previous iteration. Formally, we can say that $F(\theta^{k+1}(R \cup \{(i, j)\}), R \cup \{(i, j)\}) \geq F(\theta^k(R), R)$.

**Learning $g_\phi$.** Objective (1) is non-differentiable in $\phi$ because of the $\text{argmax}_{\boldsymbol{\beta}}\, g_\phi(\bullet)$ input to $f_\theta$ and the unknown $Z$. We first get rid of the $\text{argmax}$ term via the following surrogate:

$$\underset{\phi}{\text{argmax}} \sum_{\substack{i \in D, j \in B_i \\ \pi(\mathbf{x}_{ij})=1}} \log f_\theta(y_i \,|\, Z(z_i, \text{argmax}_{\boldsymbol{\beta}}\, g_\phi(\boldsymbol{\beta} \,|\, \mathbf{x}_{ij}, \boldsymbol{\beta}_{ij}))) $$
$$\approx \underset{\phi}{\text{argmax}} \sum_{\substack{i \in D, j \in B_i \\ \pi(\mathbf{x}_{ij})=1}} \max_{\boldsymbol{\beta}} \log\left[ f_\theta(y_i \,|\, Z(z_i, \boldsymbol{\beta}))\, g_\phi(\boldsymbol{\beta} \,|\, \mathbf{x}_{ij}, \boldsymbol{\beta}_{ij}) \right] \quad (6)$$

Next we account for the unknown $Z$ by partitioning all examples in $D$ into two groups — the set $D_\delta$ which contains groups with at least one instance where good $\boldsymbol{\beta}$s are available (*i.e.* $\max_r f_\theta(y_i|\mathbf{x}_{ir}) > 1 - \delta$), and the remaining objects $D - D_\delta$ where no good instances are available. For the instances in $D - D_\delta$ we need to find a good $\boldsymbol{\beta}$ to train $g_\phi$. For this we first estimate a function $f^{\text{CF}}(y_i|\mathbf{x}_{ij}, \boldsymbol{\beta})$ that estimates the confidence $f_\theta(y_i|Z(z_i, \boldsymbol{\beta}))$. That is, it approximates $f_\theta$ when $\boldsymbol{\beta}_{ij}$ is replaced by $\boldsymbol{\beta}$ for the $i$-th object. We estimate this quantity as the average classifier accuracy on objects with similar labels and under settings $\boldsymbol{\beta}$. In general, for continuous $y, \boldsymbol{\beta}$ this can be fit as a regression problem. For discrete $y, \boldsymbol{\beta}$, simple fractional estimates were found adequate in our experiments. We compute these estimates by defining the following counterfactual as $f^{\text{CF}}(y \,|\, \mathbf{x}, \boldsymbol{\beta}) = \frac{\sum_{(i,j) \in V} \mathbb{I}[y_i = y, \boldsymbol{\beta}_{ij} = \boldsymbol{\beta}] f_{\hat{\theta}}(y_i = y \,|\, \mathbf{x}_{ij})}{\sum_{(i,j) \in V} \mathbb{I}[y_i = y, \boldsymbol{\beta}_{ij} = \boldsymbol{\beta}]}$; where $\mathbb{I}[\bullet]$ is an indicator function and $\hat{\theta}$ is the output of Algorithm 1. With these two terms, we maximize the following objective:

$$\max_{\phi} \sum_{\substack{i \in D_\delta \\ j \in B_i}} \max_{r \in B_i} \log\left[ f_\theta(y_i \,|\, \mathbf{x}_{ir})\, g_\phi(\boldsymbol{\beta}_{ir} \,|\, \mathbf{x}_{ij}, \boldsymbol{\beta}_{ij}) \right] + \sum_{\substack{i \notin D_\delta \\ j \in B_i}} \log g_\phi\left( \text{argmax}_{\boldsymbol{\beta}}\, f^{\text{CF}}(y_i \,|\, \mathbf{x}_{ij}, \boldsymbol{\beta}) \,|\, \mathbf{x}_{ij}, \boldsymbol{\beta}_{ij} \right)$$
$$(7)$$

**Computation of $\pi$.** Our training objective (1) suggests that $\pi(\mathbf{x}_{ij}, \boldsymbol{\beta}_{ij}) = 1$ only if $f_\theta(y_i \,|\, \mathbf{x}_{ij}) < f_\theta(y_i \,|\, \mathbf{x}'_{ij} = Z(z_i, \boldsymbol{\beta}'_{ij}))$ where $\boldsymbol{\beta}'_{ij} = \text{argmax}_{\boldsymbol{\beta}}\, g_{\hat{\phi}}(\boldsymbol{\beta} \,|\, \mathbf{x}_{ij}, \boldsymbol{\beta}_{ij})$. Since the recourse budget is limited, we cannot obtain $\mathbf{x}'_{ij}$ for all instances to compute $\pi$. Therefore, in practice, we use $f^{\text{CF}}(\bullet|\mathbf{x}_{ij}, \boldsymbol{\beta}'_{ij})$ as a proxy for $f_\theta(\bullet \,|\, \mathbf{x}'_{ij} = Z(z_i, \boldsymbol{\beta}'_{ij}))$. Specifically, we set

$$\pi(\mathbf{x}_{ij}, \boldsymbol{\beta}_{ij}) = \mathbb{I}[f^{\text{CF}}(y_{\max} \,|\, \mathbf{x}_{ij}, \boldsymbol{\beta}'_{ij}) > f_{\hat{\theta}}(y_{\max} \,|\, \mathbf{x}_{ij})] \text{ where } y_{\max} = \underset{y}{\text{argmax}}\, f_{\hat{\theta}}(y \,|\, \mathbf{x}_{ij}) \quad (8)$$

We call our overall training method as RECOURSENET, which is summarized in Algorithm 3.

### 3.2 Theoretical Analysis

In this section we present the conditions on $\theta, \phi, \pi$ under which RECOURSENET will be successful in providing recourse. The proofs of the propositions are given in Appendix A.

**Proposition 1** *Assume that $Z$ is $L_{\boldsymbol{\beta}}$-Lipschitz with respect to $\boldsymbol{\beta}$, the model $\log f_\theta(y \,|\, \mathbf{x})$ is $L_x$-Lipschitz with respect to $\mathbf{x}$. Given $i \in D$ and $j \in B_i$, if the set $\text{Rec}_\Delta(\theta, \mathbf{x}_{ij}, y_i)$ is non-empty and the recourse network $g_\phi$ gives a modified $\boldsymbol{\beta}'_{ij}$ such that $||\boldsymbol{\beta}'_{ij} - \boldsymbol{\beta}|| \le \epsilon$ for some $\boldsymbol{\beta} \in \text{Rec}_\Delta(\theta, \mathbf{x}_{ij}, y_i)$, then, for $\Delta > t L_x L_{\boldsymbol{\beta}} \epsilon$ with $t > 1$ we have:*

$$\log f_\theta(y_i \,|\, Z(z_i, \boldsymbol{\beta}'_{ij})) > \log f_\theta(y_i \,|\, \mathbf{x}_{ij}) + (1 - 1/t)\, \Delta \quad (9)$$

The above proposition suggests that as long as $g_\phi(\bullet \,|\, \mathbf{x}_{ij}, \boldsymbol{\beta}_{ij})$ is close to some $\boldsymbol{\beta} \in \text{Rec}_\Delta(\theta, \mathbf{x}_{ij}, y_i)$, then the accuracy provided by the classifier $f_\theta$ improves. One of the key assumption of this proposition is the non-emptiness of $\text{Rec}_\Delta(\theta, \mathbf{x}_{ij}, y_i)$. In the following proposition, we find the requirements for such conditions in terms of the true classifier $f_{\theta^*}$.

**Proposition 2** *Let us assume that the true conditional distribution of $y$ given $\mathbf{x}$ is $f_{\theta^*}$, $\log f_\theta(y \,|\, \mathbf{x})$ is $L_\theta$-Lipschitz w.r.t. $\theta$ and $||\theta - \theta^*|| \le \delta$. Moreover, we define the following quantities: $\Delta^{(i,j)} = \max_{r \in B_i} [\log f_{\theta^*}(y_i \,|\, \mathbf{x}_{ir}) - \log f_{\theta^*}(y_i \,|\, \mathbf{x}_{ij})]$, $A = \{(i,j) \in V \,|\, \Delta^{(i,j)} > 0\}$, and $\Delta_0 = \min_{(i,j) \in A} \Delta_{i,j}$. Then, we have the following results:*

1. *For $(i,j) \in A$, $\text{Rec}_{\Delta_0}(\theta^*, \mathbf{x}_{ij}, y_i)$ is non-empty.*

2. *Given $(i,j) \in V$, if we have $\delta < \frac{\Delta_0}{2L_\theta}$, then $\text{Rec}_\Delta(\theta, \mathbf{x}_{ij}, y_i)$ is non-empty for $\Delta < \Delta_0 - 2L_\theta\delta$*

3. *If the recourse network $g_\phi$ gives us a modified $\boldsymbol{\beta}'_{ij}$ such that $||\boldsymbol{\beta}'_{ij} - \boldsymbol{\beta}|| \le \epsilon$ for some $\boldsymbol{\beta} \in \text{Rec}_\Delta(\theta, \mathbf{x}_{ij}, y_i)$ with $\Delta < \Delta_0 - 2L_\theta\delta$, then, for $\epsilon < (\Delta_0 - 2L_\theta\delta)/(tL_{\boldsymbol{\beta}}L_x)$ with $t > 1$, we have:*

$$\log f_\theta(y_i \mid \mathbf{x}_{ij}) < \log f_\theta(y_i \mid Z(z_i, \boldsymbol{\beta}'_{ij})) - (1 - 1/t)(\Delta^{(i,j)} - 2L_\theta\delta) \qquad (10)$$

# 4 Experiments

In this section, we experiment with several datasets to show that RECOURSENET's training strategy outperforms existing methods or simpler alternatives. Our experiments are designed to answer the following research questions through empirical evaluations:

1. In the training of $f_\theta$, what is the impact of subsetting the training set when compared with default alternatives like training on all available labeled data.

2. In deciding when to trigger recourse, how effective is our method, in contrast to just asking recourse on low confidence examples?

3. In training the recourse recommender, how important was it distinguish between objects with and without good $\beta$s? During inference, how important is it to make instance specific recourse recommendations instead of a single ideal beta?

We could not find any existing benchmark that records different environment settings under which objects are rendered. Thus we generate datasets that admit causal relationship across $\mathbf{x}, \boldsymbol{\beta}, z$ and $y$ as follows: we first sample a class label from the class prior $y \sim \text{Pr}(\bullet)$ and then we choose $B_i$ settings by sampling $\boldsymbol{\beta}$s drawn from a $\text{Pr}(\boldsymbol{\beta} \mid y)$. Finally we generate $\mathbf{x}$ under the $B_i$ chosen environments. We generate 4 datasets of varying complexities as shown in the Table 2.

**Shapenet-Large** Shapenet consists of three dimensional models of many kinds of objects that can be mapped into two dimensional pixel maps under various environments [3]. Each environment $\boldsymbol{\beta}$ represents the camera settings provided by (`view, zoom level, light color`). We select $|\mathcal{Y}| = 10$ classes and draw 250 objects from each class to obtain a total of $|D| = 2500$ objects. For each object, we draw $B_i = 4$ different $\boldsymbol{\beta}$s from a set of $|\mathcal{B}| = 9$ possible camera settings and render them under these settings. Among the four environments, we ensure that each $z_i$ contains a $\boldsymbol{\beta}$ that renders it properly with a probability 0.8. To make the task challenging, we corrupt the rendered $\mathbf{x}_{ij}$ using various kinds of noise from the image corruptions library[2]. In particular, we corrupt $\mathbf{x}_{ij}$ if $\boldsymbol{\beta}_{ij}$ is not a good choice for $z_i$ so as to make learning of such settings difficult for $f_\theta$.

**Shapenet-Small.** This dataset differs from Shapenet-Large in the number of environments under which each object is rendered. Among the two environments, each $z_i$ contains a good $\boldsymbol{\beta}$ with probability 0.6. This dataset is more challenging than Shapenet-Large because of scarcity in the number of objects that contain atleast one $\mathbf{x}_{ij}$ that produces good accuracy. This makes the objective (6) difficult to learn. Here also we add noise to $\mathbf{x}_{ij}$ in a manner similar to Shapenet-Large. The test set for both Shapenet-Large and Shapenet-Small is same, and contains 80 objects per class; each of them rendered under all 9 camera settings $\boldsymbol{\beta}$ thus contributing to 7200 images.

**Speech Commands Dataset.** This dataset consists of textual commands that can be converted to speech under different environments $\boldsymbol{\beta}$ defined by (`pitch, speed, noise`) sampled from $\mathcal{B}$ with $|\mathcal{B}| = 60$. We select $|\mathcal{Y}| = 20$ commonly used Alexa commands and render them to speech signals with a frame width of 0.5 seconds using Google text to speech library [3]. These speech signals are then processed into 2D mel spectrograms [27]. In particular, the training dataset consists of 2000 $z_i$ rendered under $|B_i| = 5$ environments each thereby contributing to 10000 samples. The test set contains 200 $z_i$s rendered under all 60 $\boldsymbol{\beta}$s thereby containing 12000 speech samples.

**Skin-Lesion Dataset** This dataset consists of images of skin captured using smartphone and the task is to classify among seven different skin conditions ($|\mathcal{Y}| = 7$). The dataset is taken from Kaggle [4] and we synthetically generate 9 different environments ($|\mathcal{B}| = 9$) where each environment is defined by (`zoom, illumination, contrast`). The training dataset contains 1400 objects $z_i$ rendered

---

[2]`https://github.com/bethgelab/imagecorruptions`
[3]`https://cloud.google.com/text-to-speech`
[4]`https://www.kaggle.com/code/kmader/deep-learning-skin-lesion-classification/notebook`

| Dataset | #Train objects ($|D|$) | #Renderings ($|B_i|$) | Environment ($\mathcal{B}$) | #Classes $|\mathcal{Y}|$ | #Test objects |
|---|---|---|---|---|---|
| Synthetic | 1200 | 8 | 6 dimensional bit-mask | 4 | 200 |
| Shapenet-Large | 2500 | 4 | (view, zoom level, light color) | 10 | 800 |
| Shapenet-Small | 2500 | 2 | (view, zoom level, light color) | 10 | 800 |
| Speech Commands | 2000 | 5 | (pitch, speed, noise) | 20 | 60 |
| Skin Lesion | 1400 | 4 | (zoom, illumination, contrast) | 7 | 70 |

Table 2: Summary of datasets used. The columns #Train and #Test denote the number of (latent) objects available in train, test dataset respectively. #Renderings denotes the number of environment settings under which each object $z_i$ is rendered. Environment denotes the different parameters that control the generation of $\mathbf{x}$ from $z$.

| Training Data | Shapenet-Large | Shapenet-Small | Speech-Commands | Skin-Lesion |
|---|---|---|---|---|
| Full-data (Baseline) | $71.93 \pm 0.63$ | $62.97 \pm 0.80$ | $51.85 \pm 1.08$ | $56.42 \pm 0.80$ |
| One-shot subsetting | $72.63 \pm 0.54$ | $65.55 \pm 1.11$ | $54.66 \pm 1.2$ | $60.89 \pm 1.11$ |
| Iterative greedy (Ours) | $\mathbf{77.14 \pm 0.63}$ | $\mathbf{74.13 \pm 1.10}$ | $\mathbf{65.76 \pm 1.44}$ | $\mathbf{68.62 \pm 0.90}$ |

Table 3: Comparing classification accuracy under different strategies for subsetting data for training $f_\theta$ at 100% recourse. The table shows mean $\pm$ one std. deviation of accuracies obtained over 5 seeds.

under $|B_i| = 4$ environments each and the test dataset contains 70 objects rendered under all 9 environments.

Further details about dataset preparation and results on synthetic datasets are provided in Appendix B.

**Models and Hyperparameters.** We use the same model architecture and hyper-parameters across datasets. We used Adam optimizer with default learning rate of $10^{-3}$ to optimize all our objectives. The architecture for $f_\theta$ is a Resnet18 model trained from scratch. We use budget $b = 1000$ but to avoid training the model iteratively $b$ times, we select 10 instances into $R$ at line 5 of algorithm 1 per iteration. For $g_\phi$ too we train a Resnet18 model from scratch. We obtain 512-dimensional embedding for $\boldsymbol{\beta}$ as an average of embeddings of its individual components which are trained end-to-end. We concatenate the embeddings of $\boldsymbol{\beta}$ with last layer embeddings of $\mathbf{x}$ from Resnet18 to obtain the input embedding which is then fed to a 3-layered neural network that predicts the recourse $\boldsymbol{\beta}$. We learn $g_\phi$ using the objective 7 where the $D_\delta$ is computed by sorting the minimum group loss $(min_j \mathcal{L}[(i,j)])$ and then selecting the first few groups that produce least min loss into the set $D_\delta$. The first few are chosen so that the average confidence $f_\theta(y_i|\mathbf{x}_{ij}, \boldsymbol{\beta}_{ij})$ of examples in $D_\delta$ has the maximum gap with corresponding average in $D - D_\delta$.

**RQ1: Impact of subsetting the training data in learning $f_\theta$**   We compare our iterative greedy proposal to train $f_\theta$ with two other baselines as follows:

1. **Full data:** Here we train $f_\theta$ over the entire training dataset.

2. **One-shot subsetting:** Here we subset all $b$ examples at once unlike our iterative algorithm 1. *i.e.* we compute $\mathcal{L}(i,j)$ for all samples given $\theta^0(\emptyset)$ and choose the ones that incur *top-b* values into $R$ and then maximize $F(\bullet, R)$ to obtain $\hat{\theta}$.

Table 3 shows the recourse accuracy of $f_\theta$ at 100% recourse when learned under these three different training strategies. For all three methods we use $g_\phi$ trained using our objective (7) to obtain recourse recommendations. We observe that our iterative greedy algorithm to train $f_\theta$ consistently outperforms the model trained with the entire data. This establishes the importance of training classifiers differently when recourse is an option. A classifier that is trained only on instances with 'good' environment settings is more suitable for classification under recourse, even in data hungry deep learning models. Simply subsetting by removing the worst $b$ instances is significantly worse than our iterative algorithm.

**RQ2: Evaluating our method of triggering recourse**   We compare our proposal for $f_\theta$ and $\pi$ with four other baselines and the first two are adapted from the work of [24].

1. **Score based recourse trigger:** Here, we train $f_\theta$ on entire training data. Then during inference, given a budget $b$, we seek recourse on the least $b$ confident predictions of $f_\theta$.

2. **Full automation based recourse trigger:** Here also, we train $f_\theta$ on entire training data. Then for recourse trigger, we learn an error predictor trained on the loss incurred by the classifier on training examples. During inference, for a budget $b$, we seek recourse on those examples that

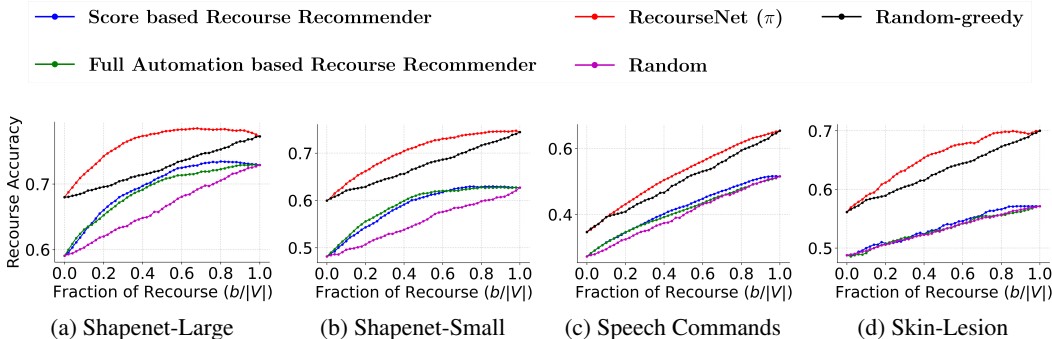

Figure 4: Variation of classification accuracy after recourse against the budget $b$, *i.e.*, the maximum number of instances selected for recourse.

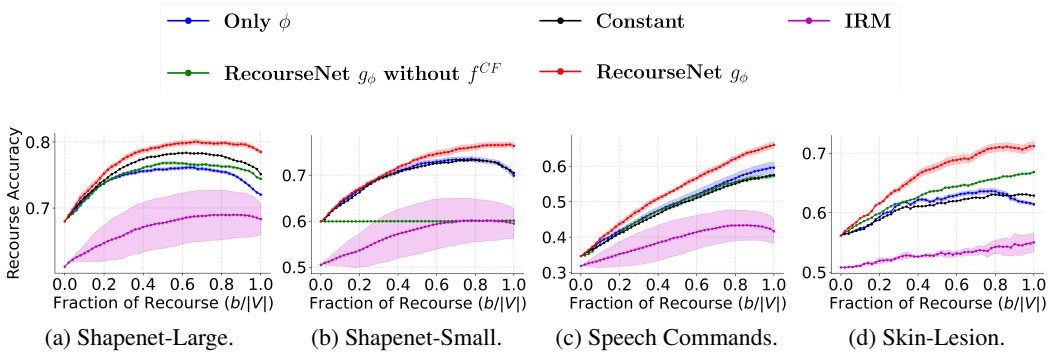

Figure 5: Variation of classification accuracy after recourse against the budget $b$, *i.e.*, the maximum number of instances selected for recourse. The figures show mean recourse accuracy $\pm$ one std. deviation obtained over five seeds.

incur the $b$ highest predicted losses. Details about the neural architecture of the error predictor is provided in Appendix C.

3. **Random trigger:** We train $f_\theta$ on entire dataset and apply recourse on instances selected randomly.

4. **Random-greedy trigger:** Here we train $f_\theta$ using our greedy algorithm 1 and then apply recourse on instances selected randomly.

Figure 4 summarizes the comparison of recourse trigger $\pi$ against the baselines. Unlike our greedy algorithm, methods that propose full training for $f_\theta$ are inferior at $0\%$ recourse. The steepness in the recourse accuracy for our proposed $\pi$ is more in comparison to other baselines because it prioritizes recourse not just the instances that suffer from poor accuracy for recourse but also the ones that respond better to recourse by means of modelling the expected recourse accuracy. Our method suggests recourse only when the expected gains that we calculate using $f^{\text{CF}}$ is positive, and performs much better than methods based purely on current classifier confidence or an estimate of the confidence. We see that both the random baselines perform much worse and follows the expected linear trend of recourse accuracy as we increase the recourse budget. These results establish the impact of our method of triggering response.

**RQ3: Evaluating training methods of recourse recommendation** $g_\phi$    We compare our $g_\phi$ against four other methods as follows.

1. **Only $\phi$:** This model takes a form similar to $g_\phi$ and learns to recourse the instances $(i, j)$ that incur top $50\%$ losses in the training data to $\boldsymbol{\beta}_{ir}$ where $r$ is obtained from $\operatorname{argmax}_r f_\theta(y_i \mid \mathbf{x}_{ir})$.

2. **RECOURSENET without $f^{\text{CF}}$:** This model trains $g_\phi$ using the objective (6).

3. **Constant:** This method emits a constant $\boldsymbol{\beta}$ recommendation independent of the features $(\mathbf{x}, \boldsymbol{\beta})$. We select constant $\boldsymbol{\beta}$ as the one that achieves the best training accuracy.

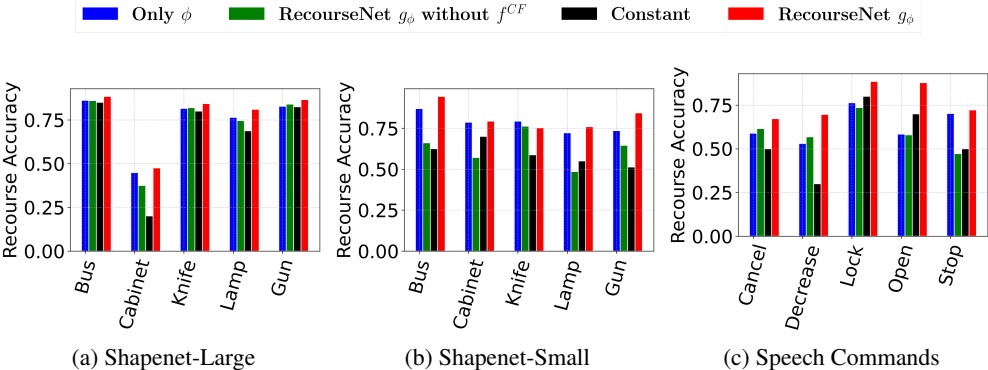

Figure 6: Accuracy of different recourse recommenders for different classes.

4. **IRM:** In this baseline, instead of $g_\phi$ we learn networks that estimate accuracy of an input $\mathbf{x}_{ij}$ on a counterfactual setting $\boldsymbol{\beta}$ using ideas from Invariant Risk Minimization literature. We extract representations of input ($\mathbf{x}$) by fine tuning a Resnet18 model with pre-trained Imagenet weights. This forms the $\Phi$ network of IRMv1 objective in [1]. The representations are multiplied with a scalar $w = 1$ and then concatenated with representation of the counterfactual environment $\boldsymbol{\beta}$ for which we want to estimate the accuracy. We use a linear layer to embed the environments ($\boldsymbol{\beta}$). The concatenated representations are then fed to a fully connected network that aims to predict the classifier's confidence ($f_\theta(y|\mathbf{x})$) on the examples. For Triage, since these methods directly model the counterfactual accuracy $P(y|\mathbf{x}, \boldsymbol{\beta}) \; \forall \boldsymbol{\beta}$, we use these predicted values in place of our prior $f^{\text{CF}}$ term in Eq. (8). The classifier is trained on full data.

The first three baselines are designed to perform an ablation study of our proposal, including assessing the importance of finding the objects that have no good $\boldsymbol{\beta}$ and thereby including them in the set $D - D_\delta$ in the $g_\phi$ objective (7). The results presented in Figure 5 shows the following observations. (1) Only $\phi$ model performs poorly on Shapenet-Large and performs on par with Constant method on other datasets. Because many groups do not have $\boldsymbol{\beta}$ that produce good accuracy, *Only $\phi$* receives noisy supervision during training. (2) RECOURSENET without $f^{\text{CF}}$ achieves a decent fit on Shapenet-Large and Specch datasets but fails miserably on the Shapenet-Small dataset. Because Shapenet-Small has $|B_i| = 2$, we can see that $50\%$ examples force the recourse recommender to predict the input $\boldsymbol{\beta}$ as is under the joint objective (6). This renders identity function as a strong local maxima which the model struggles to avoid during training. This brittleness of RECOURSENET without $f^{\text{CF}}$ to objects with no good $\boldsymbol{\beta}$ motivates the need for our current objective 7. (3) The supervision provided by the $f^{\text{CF}}$ term in our $g_\phi$ objective (7) guides instances in the set $D - D_\delta$ and thus achieves better recourse accuracy. (4) The IRM method is difficult to train as seen by the large variance, and performs poorly. This method does not sufficiently exploit the fact that the training data includes multiple views of the same object. Also, it suffers because the classifier is trained on full data. (5) One good competitor to our $g_\phi$ across the datasets is constant prediction which brings us to the other half of RQ3 – Is an instance independent constant $\boldsymbol{\beta}$ recourse recommendation always advisable?

We try to answer this question by probing the average error incurred by these methods for each class. Figure 6 summarizes the results for 5 classes which shows that unlike baselines, our $g_\phi$ garners modest to best accuracy across classes consistently. The performance of constant method in the Shapanet datasets can be attributed to the fact that many objects in it admit a unique good $\boldsymbol{\beta}$. However, this is not the case in the speech dataset because we found no one $\boldsymbol{\beta}$ to dominate in performance across classes. As a result, the recourse accuracy suffers in the speech dataset with constant $\boldsymbol{\beta}$ prediction. Thus we conclude by saying that it is always good to make instance specific recourse recommendations.

## 5 Stability of our proposals in solving the overall objective 1

In Figure 7(a) we plot the value of the overall objective 1 through the different stages of the $f_\theta$ training algorithm and in Figure 7(b) we plot for the second $g_\phi$ training phase. In both cases we observe that the value of the overall objective increases even though the two stages are not directly solving for Equation 1. This provides empirical evidence about the stability of our method.

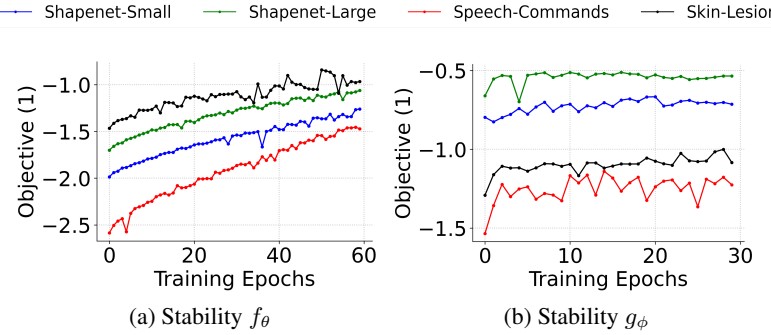

(a) Stability $f_\theta$        (b) Stability $g_\phi$

Figure 7: Stability of our proposals in solving 1

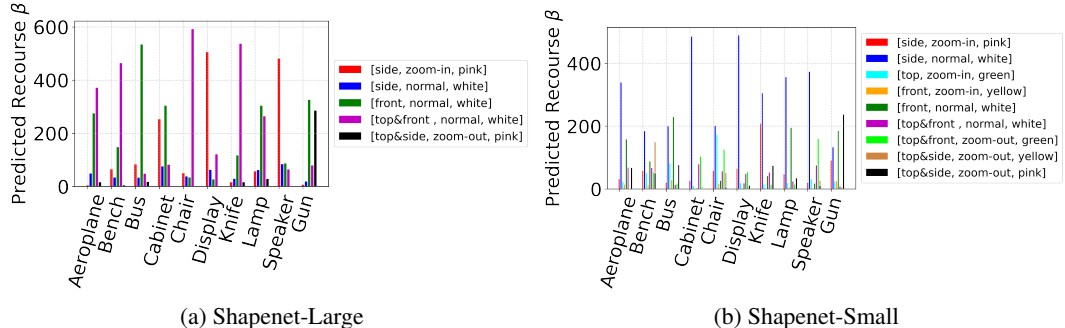

(a) Shapenet-Large        (b) Shapenet-Small

Figure 8: This figure shows the histogram of the counts of different $\beta$s predicted against each class for Shapenet-Large and Shapenet-Small datasets.

Now, we discuss about the details on how we evaluated the overall objective in these two stages. For the $f_\theta$ training phase we evaluate the objective using $R$ and $\text{Rec}_\Delta(\theta^k, \mathbf{x}_{ij}, y_i)\}$ as proxies for $\pi$ and $g_\phi$ which have not yet been trained. For the $g_\phi$ training since $\pi$ is not available, we assume full recourse and focus on the impact of $\beta$s predicted by $g_\phi$ on recourse objective.

# 6   Predicted Environments by $g_\phi$

In this experiment, we plot the counts of different $\beta$s predicted by our $g_\phi$ model against each class for Shapenet-Large and Shapenet-Small datasets. For Shapenet-Large, we see that the recourse recommender never predicted 4 out of 9 $\beta$s for any class and thus those bars are excluded from the figure 8(a). However, this is not the case for Shapenet-Small and hence all 9 $\beta$s are included.

# 7   Conclusions

In this paper, we proposed RECOURSENET that aims to make recourse recommendations to instances that are sampled from poor environments. RECOURSENET has three components: (1) classifier $f_\theta$, (2) Recourse recommender $g_\phi$ and (3) Recourse trigger $\pi$. We learn these components using a novel three level training objective without having to model the latent physical generator $Z$. Moreover, our theoretical results assure that under mild conditions, recourse is beneficial. These results in effect, press the need for recourse in order to obtain quality predictions from a model. The experiments on synthetic and real-world datasets show that our method outperforms several baselines.

Our work opens up many areas of future work. It would be interesting to extend RECOURSENET to regimes where the space of environment variable $\beta$s can be continuous. Also, multiple views of a test object collected during recourse could be exploited to improve future decisions.

# 8   Acknowledgements

Lokesh is supported by the Prime Minister Research Fellowship, Government of India.

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
