# A   Proofs of technical results

## A.1   Proof of Proposition 1

**Proposition 1** *Assume that $Z$ is $L_{\boldsymbol{\beta}}$-Lipschitz with respect to $\boldsymbol{\beta}$, the model $\log f_\theta(y \mid \mathbf{x})$ is $L_x$-Lipschitz with respect to $\mathbf{x}$. Given $i \in D$ and $j \in B_i$, if the set $\mathrm{Rec}_\Delta(\theta, \mathbf{x}_{ij}, y_i)$ is non-empty and the recourse network $g_\phi$ gives a modified $\boldsymbol{\beta}'_{ij}$ such that $||\boldsymbol{\beta}'_{ij} - \boldsymbol{\beta}|| \le \epsilon$ for some $\boldsymbol{\beta} \in \mathrm{Rec}_\Delta(\theta, \mathbf{x}_{ij}, y_i)$, then, for $\Delta > t L_x L_{\boldsymbol{\beta}} \epsilon$ with $t > 1$ we have:*

$$\log f_\theta(y_i \mid Z(z_i, \boldsymbol{\beta}'_{ij})) > \log f_\theta(y_i \mid \mathbf{x}_{ij}) + (1 - 1/t)\Delta \tag{11}$$

**Proof.** Recall that by definition in Eq. (3) in our main submission,

$$\mathrm{Rec}_\Delta(\theta, \mathbf{x}, y) = \{\boldsymbol{\beta}' \mid \log f_\theta(Z(z_i, \boldsymbol{\beta}'), y) > \log f_\theta(y \mid \mathbf{x}) + \Delta\} \tag{12}$$

Thus, for $\boldsymbol{\beta}'_{ij} \in \mathrm{Rec}_\Delta(\theta, \mathbf{x}_{ij}, \boldsymbol{\beta}_{ij})$ we have,

$$
\begin{aligned}
\log f_\theta(y_i \mid \mathbf{x}_{ij}) &< \log f_\theta(y_i \mid \mathbf{x}'_{ij} = Z(\mathbf{x}_{ij}, \boldsymbol{\beta}'_{ij})) - \Delta \\
&= \log f_\theta(y_i \mid Z(z_i, \boldsymbol{\beta})) + \log f_\theta(y_i \mid \mathbf{x}'_{ij} = Z(\mathbf{x}_{ij}, \boldsymbol{\beta}'_{ij})) \\
&\quad - \log f_\theta(y_i \mid Z(z_i, \boldsymbol{\beta})) - \Delta \\
&\overset{(1)}{<} \log f_\theta(y_i \mid Z(z_i, \boldsymbol{\beta})) + L_x ||\mathbf{x}'_{ij} - Z(z_i, \boldsymbol{\beta})|| - \Delta \\
&= \log f_\theta(y_i \mid Z(z_i, \boldsymbol{\beta})) + L_x ||Z(z_i, \boldsymbol{\beta}'_{ij}) - Z(z_i, \boldsymbol{\beta})|| - \Delta \\
&\overset{(2)}{<} \log f_\theta(y_i \mid Z(z_i, \boldsymbol{\beta})) + L_x L_{\boldsymbol{\beta}} \epsilon - \Delta \\
&\overset{(3)}{<} \log f_\theta(y_i \mid Z(z_i, \boldsymbol{\beta})) + (1/t - 1)\Delta
\end{aligned}
\tag{13}
$$

The inequality $(1)$ is due to the $L_x$ Lipschitz-continuity of $f_\theta(y \mid \mathbf{x})$ in $\mathbf{x}$. The inequality $(2)$ is due to the $L_{\boldsymbol{\beta}}$ Lipschitz-continuity of $Z(z, \boldsymbol{\beta})$ in $\boldsymbol{\beta}$. The last inequality $(3)$ follows from the assumption that $\Delta > t L_x L_{\boldsymbol{\beta}} \epsilon$.

## A.2   Proof of Proposition 2

**Proposition 2** *Let us assume that the true conditional distribution of $y$ given $\mathbf{x}$ is $f_{\theta^*}$, $\log f_\theta(y \mid \mathbf{x})$ is $L_\theta$-Lipschitz w.r.t. $\theta$ and $||\theta - \theta^*|| \le \delta$. Moreover, we define the following quantities:*

$$\Delta^{(i,j)} = \max_{r \in B_i} [\log f_{\theta^*}(y_i \mid \mathbf{x}_{ir}) - \log f_{\theta^*}(y_i \mid \mathbf{x}_{ij})] \tag{14}$$

$$A = \{(i,j) \in V \mid \Delta^{(i,j)} > 0\} \tag{15}$$

$$\Delta_0 = \min_{(i,j) \in A} \Delta_{i,j} \tag{16}$$

*Then, we have the following results:*

1. *For $(i,j) \in A$, $\mathrm{Rec}_{\Delta_0}(\theta^*, \mathbf{x}_{ij}, y_i)$ is non-empty.*

2. *Given $(i,j) \in V$, if we have $\delta < \frac{\Delta_0}{2L_\theta}$, then $\mathrm{Rec}_\Delta(\theta, \mathbf{x}_{ij}, y_i)$ is non-empty for $\Delta < \Delta_0 - 2L_\theta \delta$*

3. *If the recourse network $g_\phi$ gives us a modified $\boldsymbol{\beta}'_{ij}$ such that $||\boldsymbol{\beta}'_{ij} - \boldsymbol{\beta}|| \le \epsilon$ for some $\boldsymbol{\beta} \in \mathrm{Rec}_\Delta(\theta, \mathbf{x}_{ij}, y_i)$ with $\Delta < \Delta_0 - 2L_\theta \delta$, then, for $\epsilon < (\Delta_0 - 2L_\theta \delta)/(t L_{\boldsymbol{\beta}} L_x)$ with $t > 1$, we have:*

$$\log f_\theta(y_i \mid \mathbf{x}_{ij}) < \log f_\theta(y_i \mid Z(z_i, \boldsymbol{\beta}'_{ij})) - (1 - 1/t)(\Delta^{(i,j)} - 2L_\theta \delta) \tag{17}$$

**Proof.** The statement (1) is true by definition.

$$\log f_\theta(y_i \mid \mathbf{x}_{ij}) = \log f_{\theta^*}(y_i \mid \mathbf{x}_{ij}) + \log f_\theta(y_i \mid \mathbf{x}_{ij}) - \log f_{\theta^*}(y_i \mid \mathbf{x}_{ij}) \tag{18}$$

$$\stackrel{(1)}{\leq} \log f_{\theta^*}(y_i \mid \mathbf{x} = Z(z_i, \boldsymbol{\beta}))$$
$$+ \log f_\theta(y_i \mid \mathbf{x}_{ij}) - \log f_{\theta^*}(y_i \mid \mathbf{x}_{ij}) - \Delta_0 \tag{19}$$

$$= \log f_\theta(y_i \mid \mathbf{x} = Z(z_i, \boldsymbol{\beta}))$$
$$+ \log f_{\theta^*}(y_i \mid \mathbf{x} = Z(z_i, \boldsymbol{\beta})) - \log f_\theta(y_i \mid \mathbf{x} = Z(z_i, \boldsymbol{\beta}))$$
$$+ \log f_\theta(y_i \mid \mathbf{x}_{ij}) - \log f_{\theta^*}(y_i \mid \mathbf{x}_{ij}) - \Delta_0 \tag{20}$$

$$\leq \log f_\theta(y_i \mid \mathbf{x} = Z(z_i, \boldsymbol{\beta})) - (\Delta_0 - 2L_\theta \delta) \tag{21}$$

Thus $\mathrm{Rec}_\Delta(\theta, \mathbf{x}_{ij}, y_i)$ is non-empty for $\Delta < \Delta_0 - 2L_\theta\delta$. Next, we have

$$\log f_\theta(y_i \mid \mathbf{x} = Z(z_i, \boldsymbol{\beta})) - (\Delta_0 - 2L_\theta\delta)$$
$$= \log f_\theta(y_i \mid \mathbf{x}'_{ij} = Z(z_i, \boldsymbol{\beta}'_{ij}))$$
$$+ \log f_\theta(y_i \mid \mathbf{x} = Z(z_i, \boldsymbol{\beta})) - \log f_\theta(y_i \mid \mathbf{x}'_{ij} = Z(z_i, \boldsymbol{\beta}'_{ij})) - (\Delta_0 - 2L_\theta\delta)$$
$$\leq \log f_\theta(y_i \mid \mathbf{x}'_{ij} = Z(z_i, \boldsymbol{\beta}'_{ij})) + L_x L_\beta \epsilon - (\Delta_0 - 2L_\theta\delta) \tag{22}$$

The last inequality is due to the Lipschitzness of $f_\theta$ with respect to $\mathbf{x}$, the Lipschitzness of $Z$ with respect to $\boldsymbol{\beta}$; and, $||\boldsymbol{\beta}_{ij} - \boldsymbol{\beta}|| \leq \epsilon$.

### A.3 Analysis of our greedy algorithm

We first start with an assumption that $\log f_\theta$ is algorithmically stable, *i.e.*, if it is trained upon a dataset $V$ of size $N$, then $||\theta^*(V) - \theta^*(V')|| < \frac{\rho}{N}$, where $|V \backslash V'| = |V' \backslash V| = 1$, *i.e.*, $V$ and $V'$ has $N - 1$ elements in common and therefore, $V'$ is obtained by replacing one element of $V$. It is well known that minimizing regularized convex and $L$-Lipschitz loss functions are stable with $\rho = 2L/\lambda_{\min}$ where $\lambda_{\min}$ is the minimum eigenvalue of the regularized convex loss [26, Chapter 13, Regularization and stability]. For Polyak-Lojasiewicz (PL) loss functions with PL-coefficient $\mu$ [4, corollary 4], we have $||\theta^*(V) - \theta(V')|| < \frac{2L^2}{\mu(N-1)} \leq \frac{4L^2}{\mu N}$ for $N > 2$. Under this assumption, we state the following result:

**Proposition 3** *Suppose,* $\log f_\theta$ *is stable,* i.e., $||\theta^*(V) - \theta^*(V')|| < \frac{\rho}{N}$ *for some constant $\rho$ where $V'$ is obtained by replacing one element of $V$. Then, let us assume that the true conditional distribution of $y$ given $\mathbf{x}$ is $f_{\theta^*}$,* $\log f_\theta(y \mid \mathbf{x})$ *is $L_\theta$-Lipschitz w.r.t. $\theta$. Moreover, we define the following quantities:*

$$\Delta^{(i,j)} = \max_{r \in B_i} \left[ \log f_{\theta^*}(y_i \mid \mathbf{x}_{ir}) - \log f_{\theta^*}(y_i \mid \mathbf{x}_{ij}) \right] \tag{23}$$

$$A = \{(i,j) \in V \mid \Delta^{(i,j)} > 0\} \tag{24}$$

$$\Delta_0 = \min_{(i,j) \in A} \Delta_{i,j} \tag{25}$$

*Now, note that if $(i,j) \in A$, then it is obvious that $\mathrm{Rec}_{\Delta_0}(\theta^*, \mathbf{x}_{ij}, y_i)$ is non-empty. Assume that $|A| > b$, $||\theta(R^{(0)}) - \theta^*|| < \delta < \frac{\Delta_0}{2L_\theta}$ and $|V|$ is large enough so that $|V| > \frac{2L_\theta \rho b}{\Delta_0 - 2L_\theta\delta}$ Now if $R^{(k)}$ is solution in $R$ during the $k$-th iteration of our greedy algorithm, then the greedy algorithm will choose $(i,j)$ at each step $k \in \{1,..,b\}$ so that*

$$F(\theta^*(R^{(k)} \cup (i,j)), R^{(k)} \cup (i,j)) > F(\theta^*(R^{(k)}), R^{(k)}) \tag{26}$$

*when* $0 < \Delta < \Delta_0 - 2L_\theta \left( \delta + \frac{\rho b}{|V|} \right)$.

**Proof.** Assume that during $k$-th iteration, we have the following snapshot of the training instances:

$$V^{(k)} = \{ \underbrace{(\mathbf{x}_{i_1,j_1}, y_1), \ldots, (\mathbf{x}_{i_m,j_m}, y_m)}_{V \backslash R^{(k)}}, \underbrace{(\mathbf{x}'_{i_1,j_1}, y'_1), \ldots, (\mathbf{x}'_{i_a,j_a}, y'_a)}_{\text{Instances after applying recourse on } R^{(k)}} \} \tag{27}$$

We add *atmost* one element $(i,j)$ to $R^{(k)}$ to obtain $R^{(k+1)}$. This can be seen as replacing atmost one instance $(i,j)$ in $V$ with a new instance obtained after applying recourse on $(i,j)$. As the model is

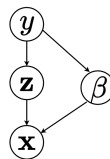

Figure 9: Causal Model that depicts the data generating process of human.

stable, then we have:

$$||\theta^*(R^{(k+1)}) - \theta^*(R^{(k)})|| \leq \frac{\rho}{|V|} \tag{28}$$

Since we start with $||\theta^*(R^0) - \theta^*|| \leq \delta$, by consecutively applying triangle inequalities, we have:

$$||\theta^*(R^k) - \theta^*|| \leq \delta + \frac{\rho k}{|V|} \leq \delta + \frac{\rho b}{|V|} \tag{29}$$

Now, from the first part of Proposition 2, we show that, whenever $\mathrm{Rec}_{\Delta_0}(\theta^*, \mathbf{x}_{ij}, y_i)$ is non-empty with $\Delta_0 > 2L_\theta \left(\delta + \frac{\rho b}{|V|}\right)$, then $\mathrm{Rec}_\Delta(\theta^*(R^{(k)}), \mathbf{x}_{ij}, y_i)$ is nonempty for $\Delta < \Delta_0 - 2L_\theta \left(\delta + \frac{\rho b}{|V|}\right)$. Hence, there will be $b$ instances for which $\mathrm{Rec}_\Delta(\theta^*(R^{(k)}), \mathbf{x}_{ij}, y_i)$ is non-empty. Now we have:

$$F(\theta^*(R^{(k)} \cup (i,j)), R^{(k)} \cup (i,j)) - F(\theta^*(R^{(k)}), R^{(k)})$$
$$= F(\theta^*(R^{(k)} \cup (i,j)), (R^{(k)} \cup (i,j))) - F(\theta^*(R^{(k)}), R^{(k)} \cup (i,j))$$
$$+ F(\theta^*(R^{(k)}), R^{(k)} \cup (i,j)) - F(\theta^*(R^{(k)}), R^{(k)}) \tag{30}$$
$$\overset{(1)}{\geq} F(\theta^*(R^{(k)}), R^{(k)} \cup (i,j)) - F(\theta^*(R^{(k)}), R^{(k)})$$

Inequality (1) is due to the fact that: $F(\theta^*(R^{(k)} \cup (i,j)), (R^{(k)} \cup (i,j))) \geq F(\theta^*(R^{(k)}), R^{(k)} \cup (i,j))$. Now given this element $(i,j)$, we will choose it for recourse if $\mathrm{Rec}_\Delta(\theta^*(R^{(k)}), \mathbf{x}_{ij}, y_i)$ is non-empty.

Now since there are at least $b$ elements for which $\mathrm{Rec}_\Delta(\theta^*(R^{(k)}), \mathbf{x}_{ij}, y_i)$ is non-empty, we will find at least $b - k$ elements which would be chosen for recourse at this $k$-th step. For those elements, we will have $\boldsymbol{\beta}_{ir} \in \mathrm{Rec}_\Delta(\theta^*(R^{(k)}), \mathbf{x}_{ij}, y_i)$ and then we have:

$$F(\theta^*(R^{(k)}), R^{(k)} \cup (i,j)) - F(\theta^*(R^{(k)}), R^{(k)}) = \log f_\theta(y_i \,|\, \mathbf{x}_{ir}) - \log f_\theta(y_i \,|\, \mathbf{x}_{ij}) > 0 \tag{31}$$

Thus, there will be at least $b - k$ elements for which

$$F(\theta^*(R^{(k)} \cup (i,j)), R^{(k)} \cup (i,j)) - F(\theta^*(R^{(k)}), R^{(k)}) > 0 \tag{32}$$

Since, we choose $(i,j)$ to be the one with highest gain, we conclude that, for any step $k \leq b$, the instance $(i,j)$ chosen for recourse, the underlying gain would be strictly positive.

## B  Additional details about experimental setup

**Causal Model.** The causal model that depicts the relationships between the variables $\mathbf{x}, \boldsymbol{\beta}, y, z$ in our dataset is shown in the Figure 9

**Synthetic Dataset.** We generate a 4 class synthetic real valued dataset with $|D| = 1200$ objects $z_i \in \mathcal{Z} = R^{d_z}$ with $d_z = 6$. The objects $z_i$ are sampled from class dependent Isotropic Gaussian distribution $\mathcal{N}(\mu_y, \Sigma_y)$ where $\Sigma_y = \mathrm{Diag}[0.1, 0.25, 0.1, 0.1, 0.25, 0.1]$ for all $y \in \mathcal{Y}$. The means $\mu_0 = [-1, 0, 0.5, 0.5, 0, 0], \mu_1 = [1, 0, 0.5, 0.5, 0, 0], \mu_2 = [0, -1, 0, 0, -0.5, -0.5], \mu_3 = [0, 1, 0, 0, -0.5, -0.5]$. Then, we draw $\boldsymbol{\beta}_{ij} \sim \mathrm{Unif}\{0,1\}^{d_z}$ such that they have exactly 3 bits set to 1 and none of them have both $\boldsymbol{\beta}_{ij}[0] = \boldsymbol{\beta}_{ij}[1] = 1$. Finally, we set $\mathbf{x}_{ij} = z_i \odot \boldsymbol{\beta}_{ij}$ for $i \in D$ and $j \in B_i$ where $|B_i| = 8$. The purpose of $g_\phi$ thus is to predict which bits in the input should be unmasked so as to make $f_\theta$ predict the correct label.

**Generating Shapenet Datasets.** As mentioned in our main submission, we work with two versions of Shapenet dataset namely Shapenet-Large and Shapenet-Small which differ in the group size $|B_i|$. While Shapenet-Large has 4 renderings for each $z_i$, Shapenet-Small has only 2 rendering for each $z_i$. Recall that we corrupt certain $\mathbf{x}_{ij}$ if $\boldsymbol{\beta}_{ij}$ used to render them is inherently noisy. Here, we expand more on how we inject noise. We use imagecorruptions python li-

| Class | front view zoom in yellow | front view normal zoom white | top view zoom in yellow | left&side zoom out pink | left&side normal zoom white | front&top normal zoom white | front&top zoom out green | side&top zoom out pink | side&front zoom out yellow |
|---|---|---|---|---|---|---|---|---|---|
| Aeroplane | ✓ | | ✓ | ✓ | ✓ | | ✓ | ✓ | ✓ |
| Bench | ✓ | ✓ | | | | | ✓ | | ✓ |
| Bus | | ✓ | | | | ✓ | ✓ | ✓ | ✓ |
| Cabinet | | | | | | | | | ✓ |
| Chair | | | | | | | | | |
| Display | | | | | | | | | |
| Knife | | ✓ | | | | | | | ✓ |
| Lamp | ✓ | | | ✓ | | ✓ | ✓ | | ✓ |
| Speaker | | | | | | | | | |
| Gun | ✓ | ✓ | ✓ | | ✓ | ✓ | ✓ | ✓ | |

Table 10: This table denotes the classes that admit noisy $\boldsymbol{\beta}$. ✓ indicates that images having the corresponding $(y, \boldsymbol{\beta})$ are corrupted w.p. $0.5$. We picked $(\boldsymbol{\beta}, y)$ pairs through visual inspection and decided to corrupt a random subset of them so as to make the learning task more challenging for $f_\theta$ thereby amplifying the need for recourse.

brary[5] for injecting noise to $\mathbf{x}_{ij}$. It provides us API for $15$ different types of noise. We selected $9$ of them namely {gaussian_noise, shot_noise, impulse_noise, frost, fog, brightness, elastic_transform, pixelate, jpeg_compression}. Each of these APIs accept an RGB image as input and outputs an RGB image with noise added to it. For each label $y$, we select a set of $\boldsymbol{\beta}$s so that any image generated under these settings $(\boldsymbol{\beta}, y)$ will be noisy with certain probability. Let us denote this set of noisy $\boldsymbol{\beta}$ for a given $y$ as $\boldsymbol{\beta}_y^{noise}$. Once we obtain $y_i, \boldsymbol{\beta}_{ij}, z_i$ following the sampling procedure depicted by the Figure 9, we render the corresponding $\mathbf{x}_{ij}$ under one of the following two cases: (a) if $\boldsymbol{\beta}_{ij} \in \boldsymbol{\beta}_{y_i}^{noise}$, we render $\mathbf{x}_{ij}$ in a noisy manner w.p. $0.5$ *i.e.* we subject the image rendered using $(\boldsymbol{\beta}_{ij}, z_i)$ to one of the 9 noises selected uniformly at random thereby rendering a noisy $\mathbf{x}_{ij}$. (b) if $\boldsymbol{\beta}_{ij} \notin \boldsymbol{\beta}_{y_i}^{noise}$, we simply render $\mathbf{x}_{ij}$ in the setting $(\boldsymbol{\beta}_{ij}, z_i)$ without adding any noise to it.

**Generating Speech Commands Dataset.** For this dataset, we choose 20 commands with $\mathcal{Y} = $ {cancel, disable, enable, decrease, increase, good morning, good night, lock, open, door, pauseplay, set, show, skip, snooze, start, stop, turn off, turn on}. We chose rhyming words so as to make the classification task harder. Unlike Shapenet, we decided to embed noise in sample generation as part of $\boldsymbol{\beta}$ itself so as to simulate real life scenarios. Because we work with Mel spectograms (images), we fixed the model architecture for $f_\theta, g_\phi$ to be the same as that of Shapenet.

**Generating Skin Lesion Dataset.** This dataset consists of images of skin captured using smartphone and the task is to predict different skin conditions ($|\mathcal{Y}| = 7$) namely {melanocytic nevi, melanoma, basal cell carcinoma, actiniv keratoses an, vascular lesions, benign keratoses lik, dermatofibroma}. The dataset is taken from from Kaggle [6] and synthetically generated environments. We generate images under 9 different environments ($|\mathcal{B}| = 9$) where each environment is defined by (zoom, illumination, contrast). For zoom, we assume that the original image is at $100\%$ zoom level and create two additional zoom levels namely $175\%, 250\%$. For illumination, we chose three values to simulate the impact of a skin image captured in light, dark, and the original image. For contrast also we chose three values and simulated low, normal and high contrast skin images. We fixed the model architecture for $f_\theta, g_\phi$ to be the same as that of Shapenet.

## C Results on Synthetic Dataset

Here, we compare the performance of various recourse trigger and recourse recommender methods on the synthetic dataset. We summarize the results in Figure 11 — we make the following observations. (1) Since the generated dataset is not linearly separable, the accuracy of $f_\theta$ is $77\%$. Moreover, the greedy algorithm for training $f_\theta$ improves the accuracy by $3\%$ over a model that trains on all data. (2) The accuracy provided by both recourse trigger $\pi$ and recourse recommender $g_\phi$ improves as we

---

[5]https://github.com/bethgelab/imagecorruptions

[6]https://www.kaggle.com/code/kmader/deep-learning-skin-lesion-classification/notebook

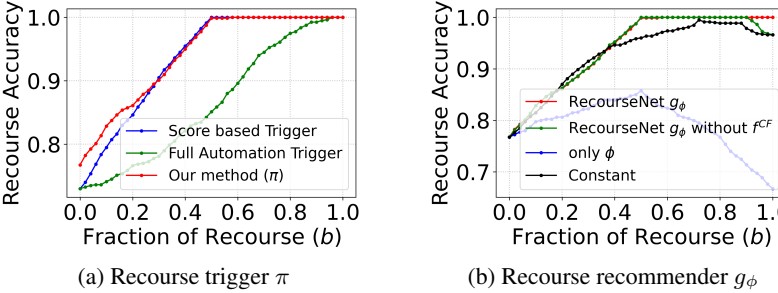

(a) Recourse trigger $\pi$      (b) Recourse recommender $g_\phi$

Figure 11: Recourse accuracy vs recourse fraction *i.e.* maximum instances that can undergo recourse for Synthetic dataset. Panel (a) shows performance comparison of recourse trigger $\pi$ with baselines. Panel (b) shows performance comparison of recourse recommender $g_\phi$ with a constant predictor.

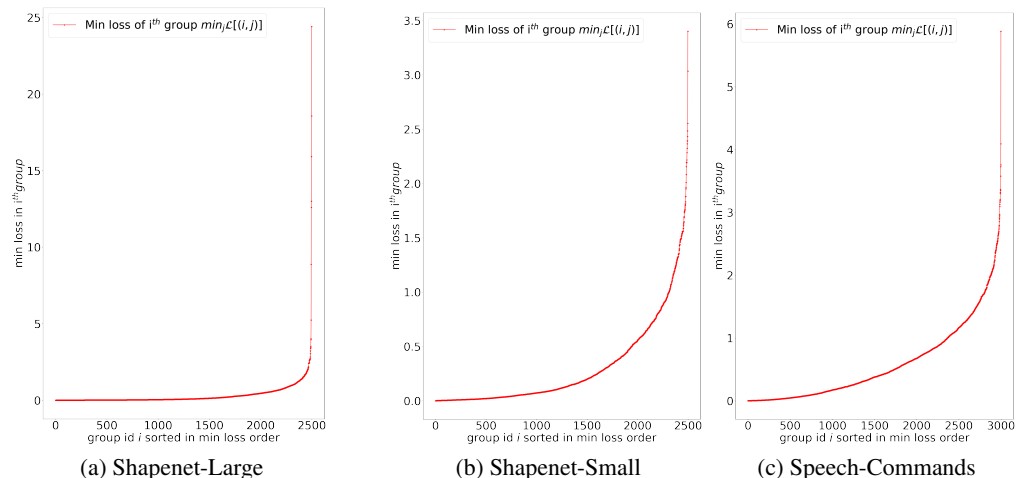

(a) Shapenet-Large      (b) Shapenet-Small      (c) Speech-Commands

Figure 12: This shows the min loss in each group in a sorted order. We use this to select the groups into $D_\delta$. As discussed in the main submission, the groups in $D_\delta$ have atleast one good feature and thus its min loss must be very close to 0. In this view, we set $D_\delta$ = the first 1800 min loss groups for Shapenet-large and the first 1250 min loss groups for shapenet-small. For Speech commands we set the first 1400 groups as part of the set $D_\delta$.

increase $b$. We notice in the dataset that it is necessary to have $1^{\text{st}}$ bit unmasked for instances labelled $\{y = 0, y = 1\}$ and $2^{\text{nd}}$ bit unmasked for the classes $\{y = 2, y = 3\}$ so that $f_\theta$ can predict them correctly. Our $g_\phi$ is able to learn this pattern using cues from the remaining bits as expected. (3) We observe a linear trend in improvement until about $48\%$; beyond which we observe a flat trend at $100\%$ recourse accuracy. This is because $\boldsymbol{\beta}$ are randomly generated which leaves us with $\approx 50\%$ bad instances that require recourse. Only $\phi$ performs poorly because of arbitration in the supervision provided by the pseudo labels that are committed while training. The model has no flexibility to pick and choose alternative good $\boldsymbol{\beta}$s in accordance with $g_\phi$ for instances where $\boldsymbol{\beta}$ prediction becomes hard. (4) Constant prediction on the other hand fails to emit instance specific recourse recommendation and hence suffers to improve the recourse accuracy consistently.

## D  Additional Baselines

We added new baselines to compare with RECOURSENET. In all these we train $f_\theta$ on the entire training dataset but instead of $g_\phi$ we learn networks that estimate accuracy of an input $\mathbf{x}_{ij}$ on a counterfactual setting $\boldsymbol{\beta}$ using ideas from the domain-invariant representations and Individual Treatment Effect estimation literature. **(1) Domain Adversarial Neural Network based training.** This method [6] aims to learn domain invariant representations using GANs based minmax objective. We extract representations of input ($\mathbf{x}$) by fine tuning a Resnet18 model with pre trained Imagenet weights. Then from the representation layer, we spawn a domain classifier that predicts the environment $\boldsymbol{\beta}$ that

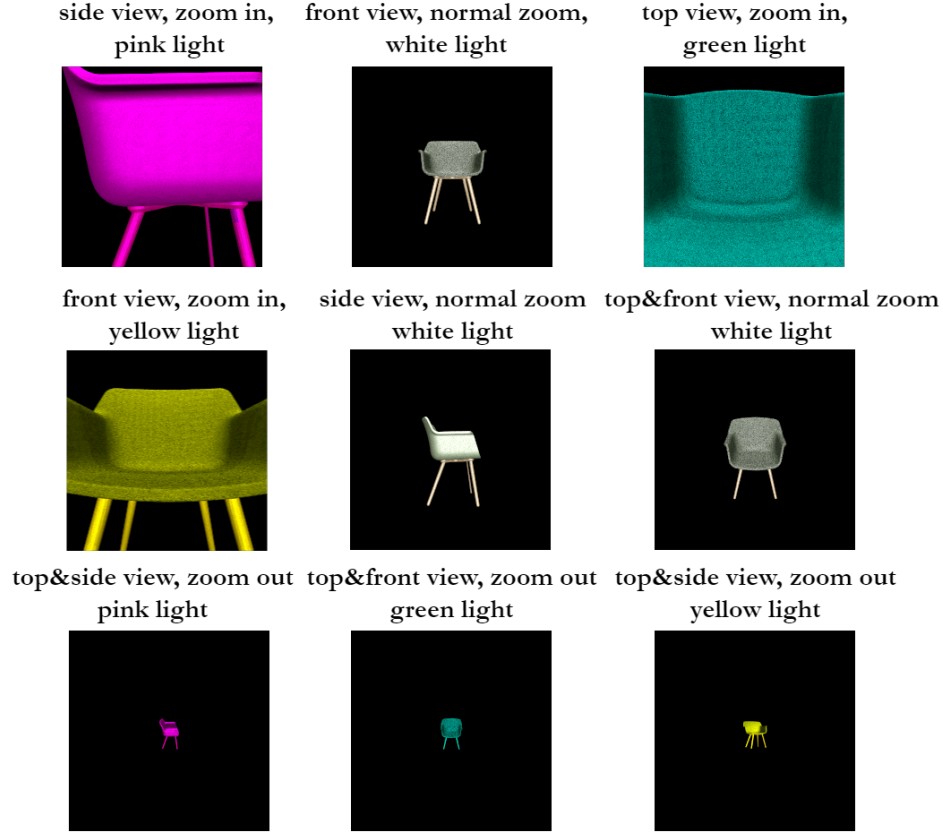

Figure 13: This figure shows renderings of a chair object under different $\boldsymbol{\beta}$s. Each $\boldsymbol{\beta}$ is a 3-tuple namely *(view, zoom-level, light color)*.

generated the instance $\mathbf{x}$. We multiply $\mathbf{x}$ representation with a domain reversal layer before feeding it to the domain classifier. The representations are concatenated with environment embedding and then fed to one more Fully connected Network that is spawned out of the representation layer. This network aims to predict classifier's confidence ($f_\theta(y|\mathbf{x})$) on the examples. **(2) TARNET.** We extract representations of input ($\mathbf{x}$) by fine tuning a Resnet18 model with pre-trained Imagenet weights. From the representation layer, we spawn $|\mathcal{B}|$ fully connected layers for each $\beta \in \mathcal{B}$. Each layer is thus responsible to predict classifier's confidence ($f_\theta(y|\mathbf{x})$) on only those instances that belong to the same environment $\beta$.

For Triage, since these methods directly model the counterfactual accuracy $P(y|\mathbf{x}, \boldsymbol{\beta})\forall\boldsymbol{\beta}$, we use these predicted values in place of our prior $f^{\text{CF}}$ term in Eq (8). The results for these baselines in shown in the Figure 14. Our proposal beats all the baselines thus establishing the supremacy of out three-stage proposal for training RECOURSENET.

## E   Illustration of original and recoursed skin images

In this experiment, we visualize the original and recoursed images for the first five images in the Skin-Lesion test dataset that require recouse as per our triage policy. The visualizations are shown in the Figure 15. The images on the left are the test images before recourse and those on the right are the corresponding images that are obtained after recourse.

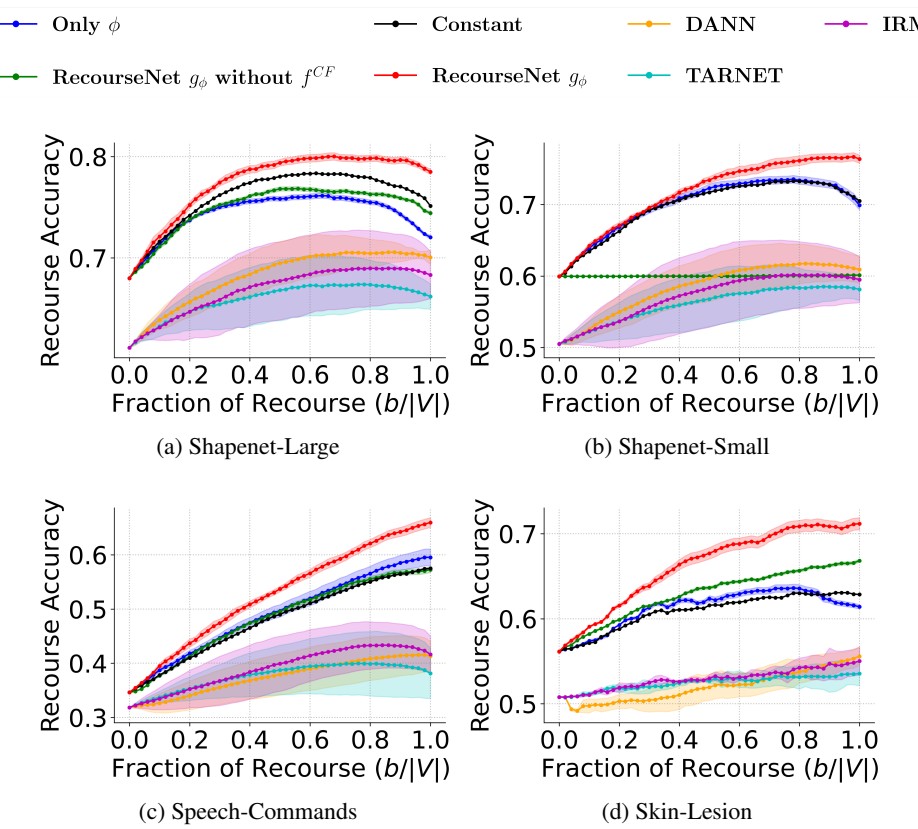

(a) Shapenet-Large

(b) Shapenet-Small

(c) Speech-Commands

(d) Skin-Lesion

Figure 14: This figure shows the performance of Recourse Recommender on all 4 datasets with newly added random baselines namely Invariant Risk Minimization, TARNET and Domain Adversarial Neural Network. The curves depict the mean Recourse accuracy $\pm$ one standard deviation over the mean for results obtained over five seeds.

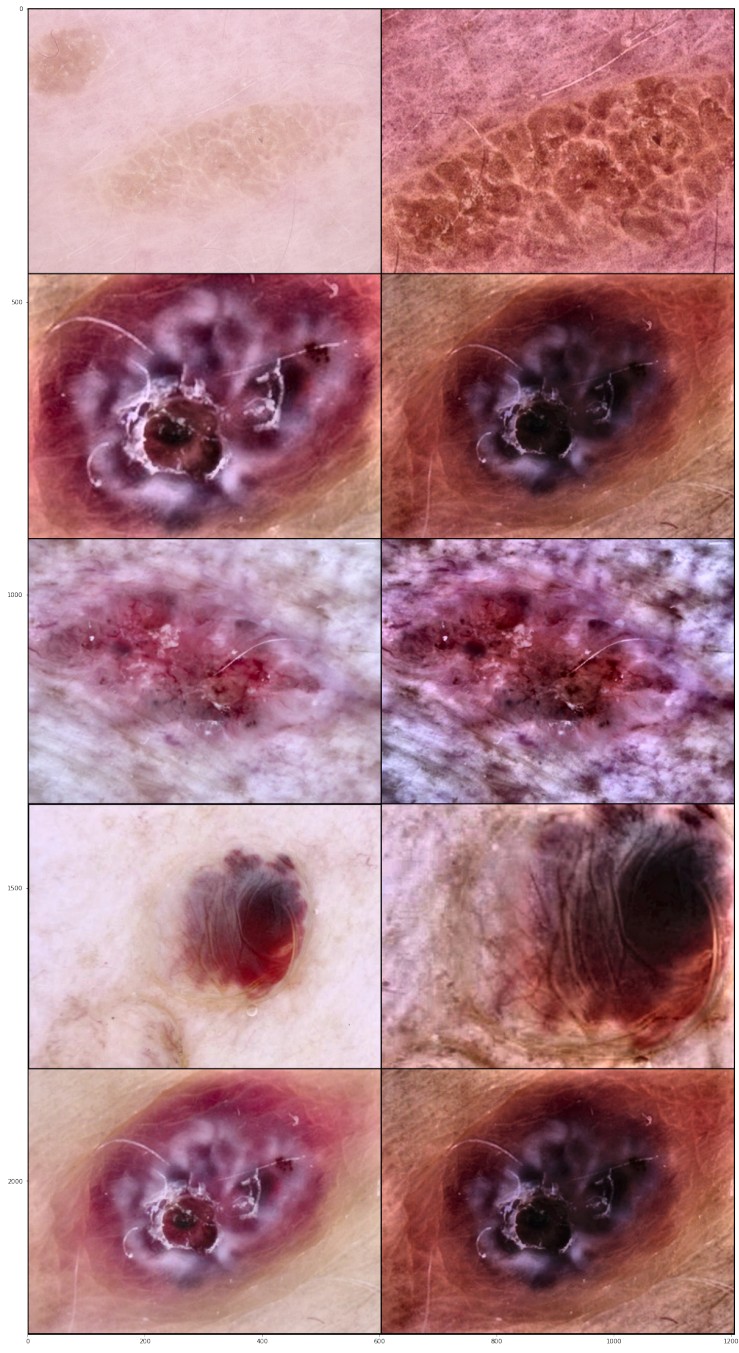

Figure 15: This figure shows the test images of the Skin-Lesion dataset before (left) and after recourse (right).