# OpenReview forum: "Learning Recourse on Instance Environment to Enhance Prediction Accuracy"
_NeurIPS.cc/2022/Conference — NeurIPS 2022 Accept_

### Official Review · Reviewer_nfUe · 2022-07-10

**Rating:** 7
**Confidence:** 3
**Soundness:** 3 good
**Presentation:** 4 excellent
**Contribution:** 3 good

**Summary:**

**Summary**: The paper tries to improve the prediction accuracy through a different perspective: recourse on instance environment. Firstly, it points out the fact that a classifier’s performance is also dependent on the data quality, that is, a good classifier is also likely to misclassify an image taken in very bad environment setting. To address this problem, the author proposes the RECOURSENET to first identify whether the data is good for classification or needed to taken from another environment setting, such as another angle. If the image is not qualified for a good classification, the recoursenet will recommend the user an appropriate environment setting. Then the recoursenet will trains a classifier to classify the image of good quality or the images generated by the user.

**Contribution**: This paper proposes a novel three-level framework for the recourse mechanism without human involvement in the training.

**Questions:**

- It would be better if the author can add a more detailed discussion about the theoretical analysis part, e.g., how does these two propositions benefit the design of the mechanism. It seems the assumption of $||\beta_{i,j}^{\prime} - \beta|| \le \epsilon$ is quite strong.

**Limitations:**

The paper's limitations are not discussed by the author.

**Strengths And Weaknesses:**

**Strengths**:
- The paper provides a novel perspective for enhancing prediction accuracy. Instead of focusing on the classification model for improving robustness, it focuses on the input data quality and introduces a recourse mechanism for improving the data quality and consequently improving the prediction accuracy. This is promising in medical applications. For example, the algorithm can be deployed on some online diagnosis platform and provides hints/tips to the user about how to take a clear image that can make the diagnosis more accurate.
- The proposed method is intuitive and relatively clearly explained.
- The experiment is thorough in terms of the ablation study, namely how every component is beneficial.

**Weaknesses**:
- The experiments are limited to small datasets.
- There is some inconsistency in terms of notation and writing. For example, the environment setting appears as $B$ or $\mathcal{B}$. And in line 88: it is inconsistent with the algorithm when saying that humans do not participate in (the) prediction task but they only generate new instances under the recommended environments.
- How these two propositions benefit seems to be unclear to me.

---

> ### Author Response · Authors · 2022-08-02
> **Reviewer nfUe Rebuttal**
>
> The environment setting appears as B or \mathcal{B}.
>
> Response: We use $B\_i$ to denote the environment settings available for the $i^{th}$ object and $\mathcal{B}$ to denote the space of all environments.
>
>  A more detailed discussion about the theoretical analysis part
>
> Response: Note that our method involves three stage training (classifier $f_{\theta}$, recourse recommender $g_{\phi}$ and recourse trigger $\pi$) rather than end–end training. During training of the first stage, we do not have access to $g_{\phi}$ which is why we resorted to the design of the recourse set $\text{Rec}_{\Delta}$. Now, the theoretical results reveal that with such a proxy design of a recourse set, it is possible to improve the classification accuracy as long as the margin $\Delta$  is higher than a threshold.  Because of these propositions, we are able to remove examples during greedy algorithms. Prior work did not consider such margin based filtering to remove examples.
>
> We would like to highlight that $\beta’_{i,j}$ is the $\beta$ predicted by $g_{\phi}$. Hence, the condition $|\beta’_{i,j} -\beta| \leq \epsilon$ reflects the quality of $g_{\phi}$. In other words, our assumption is only that $g_{\phi}$ is well trained. A well fitted model $g_{\phi}$ should provide a $\beta’_{i,j}$ which is close to *atleast* one $\beta$ in the recourse set $\text{Rec}_{\Delta}$.

---

### Official Review · Reviewer_7yCh · 2022-07-12

**Rating:** 4
**Confidence:** 3
**Soundness:** 2 fair
**Presentation:** 2 fair
**Contribution:** 2 fair

**Summary:**

The paper proposes a mechanism, termed RecourseNet,  to identify and modify input instances so as to achieve a better classification outcome from a classifier. These modifications are performed in the latent space of environment-settings.
RecourseNet consists of three components – a classifier, a recourse-trigger (which selects instances that require modification), and a recourse-recommender (that makes modifications).  A mechanism for training the three components is detailed. Conditions under which the recourse-recommendation leads to improved accuracy are studied. Experiments are performed on ShapeNet, Speech Commands dataset, and a synthetic dataset.


**Questions:**

* L21: During training, aren’t such adverse examples (if labeled correctly) important for generalization? It would be concerning if all training instances were from a fixed set of views/environments
* It is not clear to me what is meant by an ‘upstream’ classifier? In Fig 1, the classifier seems to be downstream.
* L110: D is used to index the set of objects. Maybe better to use different notation for training dataset.
* In Eq 1 (training objective), what happens if training data does not contain some objects? How is the summation over i handled?
* L126 typo: is it analytical form of Z?
* In Algorithm 1, does the TRAIN() function solve for \theta using Eq 5? If that is the case, how is Rec constructed? How is x_ir computed?
* L157: It is not clear why this is an approximation.
* L169: What is the significance of computing \pi? Doesn’t Algorithm 1 already compute \pi?


**Limitations:**

Yes

**Strengths And Weaknesses:**

STRENGTHS
==========
+ Offering recourse in the form of alternative environments is interesting, especially for images.
+ The proposed optimization problem is original, and various tricks are proposed to solve it.
+ Various ablations of RecourseNet are studied and the design choices are justified through experimental results.

WEAKNESSES
===========
- It is not clear what the problem RecourseNet actually solves. Is it trying to perform classification when training-data is noisy?
- L21: During training, aren’t such adverse examples (if labeled correctly) important for generalization? It would be concerning if all training instances were from a fixed set of views/environments. Finding alternative environments makes sense during test time, because the classifier might not have seen such examples during training.
- Misuse of the concept of offering recourse: In various places (L5,7,37) the problem is defined as “finding an alternative environment setting so that the classifier is more likely to get a correct prediction”. In traditional recourse, a fixed  classifier is assumed, and actionable changes are suggested to a user (input instance) that has been negatively classified. I am not convinced that performing pre-classification data modifications can be termed as recourse. The usage of the concept of recourse needs justification.
- Lack of a suitable baseline: In experiments, the questions studied are for different settings of the proposed algorithm. For instance, Section 4.1 is an ablation study that measures how important their iterative greedy proposal is for solving eq (5). It would be interesting to see how RecourseNet performs when compared to other methods that solve the same problem.  (Because the underlying problem it solves is not clear to me, it becomes difficult for me to recommend baselines at this time, but I suppose baselines like Invariant Risk Minimization or other methods that attempt to find domain-invariant representations could be baselines.)

---

> ### Author Response · Authors · 2022-08-02
> **Reviewer 7yCh Rebuttal**
>
> It is not clear what the problem RecourseNet actually solves. Is it trying to perform classification when training-data is noisy?
>
> No, there seems to be a misunderstanding.  In addition to training the classifier,  our goal is also to train networks $g_\phi$ and $\pi$ that for *new* objects (unseen during training) can recommend if recourse will be useful, and  recommend a new setting that could lead to better accuracy on the object.
>
> Aren’t such adverse examples (if labeled correctly) important for generalization? …
>
> Response: Unlike baselines which remove adverse examples, we carefully only remove those examples for which there is a counterpart good example.  This step ensures that the classifier is trained to be optimal on the instances that will be submitted to it *after* recourse.   In some data-hungry models, it might be useful to pre-train the classifier on all examples.  However, if we want the classifier to be optimized for the test scenario (under recourse), we recommend fine-tuning with good examples like in our method.
>
>
>
>
> More Baselines:
>
> Response: We have added all the results in Section H of our revised Appendix.
>
> ‘upstream’ classifier
>
> Response: corrected it. Thanks.
>
> what happens if training data does not contain some objects? …
>
> Response: Our training data contains a sample from a latent distribution of objects and is designed to generalize to new objects during test time. Please note that we design  $g_\phi$ (Eq 8) and $\pi$ (Eq 9) in subsequent stages so that we can determine the recourse action on new test objects.
>
> is it analytical form of Z?
>
> Response: Yes, instead of analytical form of $\beta$, it should be read as Z.
>
> In Algorithm 1, does the TRAIN() function solve for \theta using Eq 5?...
>
> Response: Yes, the TRAIN function solves $\theta$ for *a fixed recourse set* $Rec_{\Delta}$. $Rec_{\Delta}$ is computed using Eq. (5). Note that by definition, its elements $\beta’$  corresponds to those instances which already exist in the training set. Specifically, given an instance $(x, \beta, y)$,  it consists of those $\beta’$ where $(x’, \beta’, y’)$ exists in the training set and improves the log-likelihood by a margin $\Delta$.
>
> L157: It is not clear why this is an approximation.
>
> Response: It is a typo: $\max_{\phi}$ should be replaced as $\text{argmax}_{\phi}$. Please see the fixed formula in the revised version.
>
> L169: What is the significance of computing \pi? …
>
> Response: Note that $\pi$ should depend on the quality of classification after recourse, if an instance is selected for recourse. Therefore, it should depend strongly on the output of $g_{\phi}$ (L169–173). Now, we adopt a  three-stage training instead of end–end training.  Algorithm 1 is the first stage of it and thus it does not  have access to the output of $g_{\phi}$ since the latter is trained in the next stage. Thus, Algorithm 1 uses *a proxy set of instances* which could undergo recourse ($\text{Rec}_{\Delta}$) to train $\theta$. This proxy set is  explicitly enumerated on the training data and does not generalize to the test data.  The computation of $\pi$ in L169 is what decides recourse on the test data.

---

> > ### Comment · Reviewer_7yCh · 2022-08-09
> > **Response to rebuttal**
> >
> > Thank you for your responses, and the additional results.
> >
> > > *No, there seems to be a misunderstanding. In addition to training the classifier, our goal is also to train networks  and  that for new objects (unseen during training) can recommend if recourse will be useful, and recommend a new setting that could lead to better accuracy on the object.*
> >
> > -> This question is actually tied to the third one, for which there was no response. The traditional definition of algorithmic recourse is that the model remains the same, and data is adapted to get a favorable decision with the same model. Considering that "recourse" is being used here for model training itself, I am not sure if this is "algorithmic recourse" as claimed. It may be important to make this distinction. Now, going with this thought that the proposed setting is different from standard algorithmic recourse, it appears that the objective is about improving training when there is "noisy" or "non-conducive" data? Separating this from traditional recourse, in my opinion, may be important for readers in the community. To me, this is an important concern since the paper is premised on recourse.
> >
> > Continuing with the above thought, it appears that the core idea of the work is actually a generative augmentation method based on latent variables that is aimed at better generalization (since the generated samples are used for training). If this is agreed, how would this method compare with standard augmentation methods (e.g. Mixup, Domain Mixup, etc) on these datasets? I think these baselines may also become necessary. I'd like to add that I do not deny that there is merit in the work -- the work is interesting and relevant too. I am just not completely convinced about its current positioning, and hence its baselines.
> >
> > -> I agree with reviewer 86ep's point on the random and random-greedy baselines too. Thank you for providing these results in Fig 11. What overhead (time and compute complexity-wise) is required to get this improvement over such baselines? I think knowing this trade-off  is important too.
> >
> > > *It is a typo: $\max_{\phi}$ should be replaced as $\arg\max_{\phi}$. Please see the fixed formula in the revised version.*
> >
> > -> Even with this, it is not clear to me how this is an approximation. Could you please help clarify?
> >
> > > *The computation of $\pi$ in L169 is what decides recourse on the test data.*
> >
> > -> I am a bit confused. Is the $\pi$ step a part of model training phase, or is it something that is done on test data? This was not stated in the paper. If this is in the test phase, how is this related to test-time training? Related references are below:
> >
> > Test-Time Training with Self-Supervision for Generalization under Distribution Shifts, ICML 2020
> >
> > Tent: Fully Test-time Adaptation by Entropy Minimization, ICLR 2021

---

> > > ### Author Response · Authors · 2022-08-09
> > > **Response to Reviewer 7yCh**
> > >
> > > On using recourse term...
> > >
> > > Response: Traditional methods attempt to recourse by finding perturbation in the input space that in turn encourages the classifier to predict the correct label. These perturbations are mostly limited to additive transformations [9, 13, 29]. We offer recourse in the environment space and let the latent physical process $Z$ generate instances. $Z$ intrinsically encapsulates the structural causal model that generates instances and our environment recommendations can be thought of as exogenous inputs to it.
> > >
> > > human-in-the-loop augmentation method...
> > >
> > > Response: We do not compare with Test-time augmentation methods because 1. We do not have gold labels during test-time, and 2.  Our focus is not to enhance accuracy of future objects based on unlabeled views of a different object.  However, we agree that the latter could be an interesting direction for future work.
> > >
> > >
> > > Is the $\pi$ step a part of model training phase...
> > >
> > > Response: $\pi$ step is done during inference. We rewrote the RecourseNet pseudocode in the revision to clearly separate out the Training (Alg 2) and Inference (Alg 3) steps.  We request the reviewer to go over the revised pseudocode.  We thank you for this question because it helped us improve the presentation of our algorithm significantly.
> > >
> > > Clarification on $\phi$ training ...
> > >
> > > Response: Let’s assume that  $g_{\phi}$  has a unit mass on $\beta_{\max} = \arg \max_{\beta'} g_{\phi} (\beta' | x, \beta)$. Under this condition,  the RHS and LHS of Eq. (6) are exactly the same. Motivated by this, we first relax the LHS using the following approximation:
> > >
> > > $f_{\theta} (y | Z(z, \beta_{\max}) ) \approx   f_{\theta} (y | Z(z, \beta_{\max}) ) g_{\phi} (\beta_{\max} | x, \beta) $
> > >
> > > Next, we further relax the above quantity as $ \max_{\beta’} f_{\theta} (y | Z(z, \beta’) ) g_{\phi} (\beta’ | x, \beta) $.
> > >
> > > Perhaps it is appropriate to call RHS as a surrogate rather than an approximation of LHS.

---

### Official Review · Reviewer_86ep · 2022-07-13

**Rating:** 5
**Confidence:** 4
**Soundness:** 2 fair
**Presentation:** 2 fair
**Contribution:** 2 fair

**Summary:**

The authors propose using a recourse mechanism that allows a model to recommend to its human users how they could improve the predictions of the model by capturing another sample of the current instance with somewhat different view hyper-parameters (less zoom, more brightness etc). The model is made of three stages, one for prediction of a target class, one for predicting whether resampling could be useful, and another recommending how the resampling should be made.

The method requires that the model has access to different views of various instances and a corresponding real value that describes somewhat the transform variation over those views.

The authors state that this could be useful in settings such as the medical setting.

They then evaluate their model on a variety of synthetic and non synthetic datasets and show that their method can improve performance successfully on those.

**Questions:**

1. The authors state that this could be useful in medical settings. But is it really? It can be expensive to get additional MRI/CT etc images for the patient and the hospital, and, more than that, for the model to work it would need to be trained on lots of variations of the same instances with certain imaging parameters annotated. Furthermore, it would need to somehow be able to generalize on different humans and hospital machines and how they interpret the recommended changes for the resampling. It simply seems too expensive for what appears to be highly uncertain and unverifiable (at in-the-wild application) predictive improvements.
2. Have you evaluated the model on randomly recoursed instances? That could help evaluate whether a. the recourse predictor is useful and b. whether the recourse recommender makes useful recommendations.
3. Since samples are being resampled, have you considered comparing your method with existing few-shot learning models? ProtoNets and Matching Nets could be an initial baseline to compare to. As sample size increases, the predictions of those models improves as well.

**Ethics Review Area:**

["I don’t know"]

**Limitations:**

The technical limitations I stated earlier stand, but in terms of negative societal impact, I don't see any obvious problems (other than potentially increased hospital spending from patients and the government alike if implemented poorly).

**Strengths And Weaknesses:**

One of my main concerns is around the idea of recourse/resampling. The authors state that this could be useful in medical settings. But is it really? It can be expensive to get additional MRI/CT etc images for the patient and the hospital, and, more than that, for the model to work it would need to be trained on lots of variations of the same instances with certain imaging parameters annotated. Furthermore, it would need to somehow be able to generalize on different humans and hospital machines and how they interpret the recommended changes for the resampling. It simply seems too expensive for what appears to be highly uncertain and unverifiable (at in-the-wild application) predictive improvements.

Originality:

The work attempts to do something novel and potentially useful. I consider it's novelty at acceptable levels.

Quality:
The paper is written well, and presented well, however the experimental quality is lacking as the model is only evaluated on datasets that are very precise in how a particular instance is viewed in terms of the view parameters given. In medical settings, for example, this would not be very possible with current datasets. The authors also do not evaluate on a more real world dataset of the types that they mentioned their method could be useful in (such as medical settings).

The authors also do not compare with randomly recoursed/resampled instances to evaluate the effectiveness of their recommender model.

Clarity:

The paper is overall quite clear, and the methods clearly described.

The paper is also lacking optimizer settings, data augmentations and regularization details for the models that were trained.

Significance:

Assuming the work could be reframed in a way that it requires a more realistic data source with more noisy imaging parameters, the idea of recourse could be a useful one. As it currently stands however, it does not seem to offer real world significance, but does offer fair research novelty and significance

---

> ### Author Response · Authors · 2022-08-02
> **Reviewer 86ep Rebuttal**
>
> Main concerns is around the idea of recourse/resampling …
>
> Response:  We agree with the reviewer that in health critical domains such as in MRI imaging where resampling is prohibitive, our method as is, is not applicable. But we aim to target healthcare on the edge  where users would be guided to tweak the settings of the environment that control the way user’s health parameters are captured using low-cost devices [21, 22]. For example, in skin-lesion detection, users can be prompted to adjust the zoom, lighting and other settings of the imaging device used to capture the users’ features. Another example is  covid detection from cough samples where  users can be prompted to adjust for volume, pace etc. to enhance accuracy of diagnosis. To elucidate the above, we have reported results on a new medical imaging dataset on skin-lesion detection. In this dataset too, we observe 13.8\% improvement in accuracy of diagnosis with recourse, thus emphasizing the need for recourse for health care applications on the edge.
>
>
> The authors also do not compare with randomly recoursed …
>
> Response: We have added the results in Figure 11 for all four datasets. We select instances randomly for recourse and show them as baselines ‘Random’ and ‘Random-greedy’. In Random, we use the classifier obtained by training on the entire dataset and in random-greedy, we use  the classifier emitted by our greedy algorithm (1). We observed that these baselines admit a linear trend in terms of recourse performance and thus fail to prioritize instances under poor environments for recourse.
>
>
> The paper is also lacking optimizer settings, …
>
> Response: We had included these details in Section E of our Appendix. We do not use any regularization on the model weights.
>
>
> It does not seem to offer real world significance …
>
> Please see response above where we discuss a new skin lesion dataset motivated from the medical domain.
>
>  Have you considered comparing your method with existing few-shot learning models ...
>
> Response: We do not do resampling and we believe that few-shot methods are not relevant for our work.

---

> > ### Comment · Reviewer_86ep · 2022-08-07
> > **Response to rebuttal**
> >
> > >Response: We agree with the reviewer that in health critical domains such as in MRI imaging where resampling is prohibitive, our method >as is, is not applicable. But we aim to target healthcare on the edge where users would be guided to tweak the settings of the >environment that control the way user’s health parameters are captured using low-cost devices [21, 22]. For example, in skin-lesion >detection, users can be prompted to adjust the zoom, lighting and other settings of the imaging device used to capture the users’ >features. Another example is covid detection from cough samples where users can be prompted to adjust for volume, pace etc. to >enhance accuracy of diagnosis. To elucidate the above, we have reported results on a new medical imaging dataset on skin-lesion >detection. In this dataset too, we observe 13.8% improvement in accuracy of diagnosis with recourse, thus emphasizing the need for >recourse for health care applications on the edge.
> >
> > Fair enough. I agree that on-the-edge applications may benefit from such a recourse recommendation network. Please ensure that you clearly explain this in the paper, as one of the motivations perhaps.
> >
> > >Response: We have added the results in Figure 11 for all four datasets. We select instances randomly for recourse and show them as >baselines ‘Random’ and ‘Random-greedy’. In Random, we use the classifier obtained by training on the entire dataset and in random->greedy, we use the classifier emitted by our greedy algorithm (1). We observed that these baselines admit a linear trend in terms of >recourse performance and thus fail to prioritize instances under poor environments for recourse.
> >
> > Ok great. This figure should be in the main paper I think. It seems to showcase key information that is very important to your conclusions.
> >
> > >Response: We had included these details in Section E of our Appendix. We do not use any regularization on the model weights.
> >
> > Generally, crucial information needed to reproduce a piece of work should be in the main body, and not the appendix. Furthermore, in the main body, you do not state the model architectures and hyperparameters used for your various models. Could I please request that such key information is added to your experiments section? Under model architectures and training details?
> >
> > >Response: We do not do resampling and we believe that few-shot methods are not relevant for our work.
> >
> > Right. Let me explain a bit further, to try to persuade you that few-shot learning methods are, in fact, quite relevant here.
> >
> > Every time you ask for recourse, you get a new view of a particular instance. That view could be used in the way that you do, which is, pass it through the network again to get a (hopefully) better prediction, or, instead, you could treat any additional views you get from recourse as a few-shot learning dataset. Methods like protonets and matching nets could integrate all views (including the 'bad' ones) into a knowledge base from which they can make better predictions. Since these models are trained for few-shot (one to five shot generally) cases, they are ideal for extracting lots of value out of little data. I was simply asking whether you considered applying some of those methods, on top of what you already do, which is to pass the new sample through the network.
> >
> > I'll raise my score a bit, but in order to accept this paper I require that the main training details and the figure showcasing your method vs random recourse is moved in the main paper.

---

> > > ### Author Response · Authors · 2022-08-09
> > > **Response to Reviewer 86ep**
> > >
> > > on the few-shot baselines ...
> > >
> > > Response: The few-shot methods are not directly applicable in our work because we work with a fixed set of labels during both training and test time, and we do not have access to gold labels during testing. However, objects obtained under different views at test time could be useful in predicting the labels for forthcoming instances in an online manner. But since gold labels of objects are not available during deployment, exact methods of exploiting this information require further investigation and we leave it up for future work.
> > >
> > > We have incorporated other suggestions in our revised paper.

---

### Official Review · Reviewer_ERbB · 2022-07-16

**Rating:** 6
**Confidence:** 3
**Soundness:** 3 good
**Presentation:** 3 good
**Contribution:** 3 good

**Summary:**

This paper proposes a RecourseNet, which learns to instance-dependently recourse to environments so that the learner gets better samples at training time and the predictor is well informed which instance to recourse with which environments. Towards this goal, they propose a novel learning objective to optimize trigger, and the parameters of predictor and recourse recommender. Due to its non-differentiability, they resort to simple heuristics to approximate the learning objective and develop an efficient learning framework. The experimental results demonstrate the efficacy of RecourseNet over simple baselines, achieving superior recourse accuracy versus fraction of recourse. The results also show that instance-wise recourse is important as well.

**Questions:**

# Questions
- How did you choose the hyperparameters $b$, $\Delta$ and $\delta$?
- In Figure 4, how did you select the 5 classes?
- In L154, I think $F(\theta^{k+1}(R\cup \{(i,j)\})) \geq F(\theta^{k}(R\cup \{(i,j)\}))$ is wrong (also, $F$ should take two arguments as an input) because we cannot guarantee the predictor performance at the next iteration for many reasons (e.g. non-convexity). What we can guarantee is $F(\theta^k, R\cup \{(i,j)\}) \geq F(\theta^k, R)$. Am I understanding correctly?

# Typo
- In Eq.(2), $\pi(x_{ij})$ should be $\pi(x_{ij},\beta_{ij})$
- L124, 125: $(1-\pi(\cdot,\cdot)f_\theta(\cdot,\cdot))$ should be $(1-\pi(\cdot,\cdot) \log f_\theta(\cdot,\cdot))$
- In Eq.(6), in the first line there should be $\log$ before $f_\theta$ (in Appendix A, the authors say that the second line of Eq.(6) should be argmax, but it sounds weird because argmax returns $\beta$ which is not something that we maximize)
- I wonder whether the second line of Eq.(8) is correct?


**Limitations:**

The authors did not particularly mention the limitations of this work.

**Strengths And Weaknesses:**

# Strengths
- Paper is generally well written
- The motivation is very clear, and the problem setup is very interesting, novel, and practical, which I think of as the biggest contribution of this paper. The assumption that the underlying true process $Z$ is unavailable such that we need to instance-wisely recommend alternative $\beta$ sounds very reasonable.
- The learning objective (1) and (2) seems novel and looks interesting.
- The experimental results demonstrate that the proposed RecourseNet clearly outperforms the baselines (although no confidence intervals are provided and those baselines seem rather weak)
- The authors made a good effort in developing novel datasets, which can be used by other papers.

# Weaknesses

- Doubt about the original learning objective (1) and (2). I understand what (1) and (2) are meant to do, but in my understanding they can have a trivial solution: for instance, with sufficient network capacity and enough training time, the learner may simply find a trivial solution $\pi(x,\beta)=0 \forall (x,\beta)$ and simply maximize $\log f_\theta (y|x)$ as much as it can. I understand that during training and with limited computational capacity of $f_\theta$, sometimes $\pi(\cdot,\cdot)$ will become $1$ because "bad" $x$'s will have high prediction errors, but it is only in terms of training process, not encoded by the learning objective itself. In other words, the learning objective itself cannot distinguish between bad local optima (trivial solutions) and good local optima (meaningful solutions) if those minima can achieve the same loss value. So I guess we need more discussion about whether (1) and (2) are well-posed in the first place.

- Crude approximation of the learning objective. I understand the difficulty of dealing with the original learning objective (1) and (2) and crude approximations are inevitable. Then, what we need to empirically show is that the proposed approximations are indeed maximizing the original objective (1) and (2) in a stable manner. I would appreciate if the authors could show this in the rebuttal phase.

- Typos spread in the text, especially in the mathematical notations. (See questions and typos below)

- Limited baselines. I understand that there may not exist suitable baselines to compare because the problem setup is novel, but I felt they are too few. As an example, in Figure 2 the authors could have added more baselines that train $f_\theta$ selectively, in order to truly show the validity of the "steepness" of the red curves. (i.e. the different steepnesses may come from the superiority of $f_\theta$?) Honestly I'm not an expert in this area, so I will have discussions about the possible baselines with other reviewers.

- In Figure 4, it would be more informative if the authors could actually show the different $\beta$ realized for each class (either qualitatively or quantitatively), instead of the actual recourse accuracies.

- The experimental results do not provide confidence intervals, so the authors cannot argue statistical significancy.

# Summary
In summary, I found that the problem setup is very meaningful, interesting, novel, and practical. But the overall quality of the paper is a bit low and more experiments seem to be needed. If the authors can address my concerns, I can increase my score.

---

> ### Author Response · Authors · 2022-08-02
> **Reviewer ERbB Rebuttal**
>
> Doubt about the original learning objective (1) and (2) …
>
> Response: Indeed, if the network has such high capacity that it easily overfits on the training data, the trivial solution you mention could arise.  However, we expect mechanisms like early stopping using a validation dataset to prevent such overfitting.  Much recent work on training models for deferring to human experts make similar assumptions about the network being trained [11, 17, 21].
>
> Crude approximation of the learning objective …
>
> Response: In Figure 9(a) we plot the value of the overall objective (1) through the different stages of the $f\_\theta$ training algorithm and in Figure 9(b) we plot for the second $g\_\phi$ training phase. In both cases we observe that the value of the overall objective increases even though the two stages are not directly solving for Equation (1). This provides some indirect evidence about the stability of our method.
>
> Typos spread in the text ...
>
> Response: We have fixed the minor concerns including the typos in the revised version of the paper.
>
> Eq. (6) should be argmax…
>
> Response: Corrected it. Thanks for noticing this embarrassing typo.
>
> I wonder whether the second line of Eq. (8) is correct? …
>
> Response: This is correct.  The $f^{\text{CF}}$is already trained, which we use to find the best $\beta$ via the argmax.  $g\_\phi$ is trained to predict that argmax $\beta$ for instances which are not paired with another instance with a good beta (not part of $D\_\delta$).
>
> Limited baselines…
>
> Response: We have added more baselines and a new dataset. Please refer to our general comments.
>
> In Figure 4, …
>
> Response: We show a histogram of recourse betas predicted by our $g\_\phi$ model for Shapenet-Large and Shapenet-Small dataset. Please refer to Section G for more details.
>
> How did you choose the hyperparameters, ...
>
> Response: We set the budget as $b=1000$ for all datasets. The exact value of $\Delta$
> depends on how much improvement a user wishes to have from recourse. Specifically, if one expects to have $ a\% $ improvement after recourse, then $\Delta = \log(1+a/100)$. In our work, we set $a = 10$ and thus $\Delta = \log (1.1)$
>
> In Figure 4, how did you select the 5 classes?
>
> Response: To avoid clutter in the plots, we chose classes that better elucidate the weakness of the constant baseline. Nonetheless, we observed that our proposed method was never performing the least across classes in all the datasets.
>
> In L154, I think is wrong
>
> Response: Yes. Thanks for catching the mistake.

---

> > ### Comment · Reviewer_ERbB · 2022-08-02
> > **Thanks for the detailed comments.**
> >
> > I read the revised paper and appendix, and found them satisfactory. Thus I increase my score. I especially appreciate that the authors added more baselines and provided empirical validation that the proposed approximation indeed maximize the learning objective (1). I will further adjust my score during the upcoming discussion phase.
> >
> > The major reason I'm reluctant to further increase my score is because 1) I cannot have strong confidence on the technical soundness because authors made quite a lot of mistakes in the original submission, and 2) there are no confidence intervals in all the experiments.
> >
> > Thank you!

---

> > > ### Author Response · Authors · 2022-08-04
> > > **Thanks for the Rebuttal response.**
> > >
> > > Please find our results with confidence intervals obtained over $5$ seeds. These tables show the mean performance $\pm$ one standard deviation over the mean.
> > >
> > > The results for **$f_\theta$ training** are added in the following table
> > >
> > > | Training Data      | Shapenet-Large | Shapenet-Small     | Speech-Commands | Skin-Lesion |
> > > | :---        | :----   |          :--- | :--- | :--- |
> > > | Full-Data (Baseline)| $71.93 \pm 0.63$ | $62.97 \pm 0.80$| $51.85 \pm 1.08$| $56.42 \pm 0.80$ |
> > > | one-shot subsetting| $72.63 \pm 0.54$ | $65.55 \pm 1.11$| $54.66 \pm 1.2$| $60.89 \pm 1.11$ |
> > > | Iterative greedy (ours) | $77.14 \pm 0.63$ | $74.13 \pm 1.10$| $65.76 \pm 1.44$| $68.62 \pm 0.90$ |
> > >
> > > Next, we fix the downstream classifier and then repeat $g_\phi$ training for $5$ seeds and compute the confidence interval on them. These tables show the accuracy of recourse at two different recourse budgets. We do not include constant baseline as it does not involve training.
> > >
> > >
> > > **$25$% Recourse:**
> > >  |  Method     |  Shapenet-Large  |  Shapenet-Small      |  Speech-Commands  |  Skin-Lesion  |
> > >  |  :---         |  :----    |           :---  |  :---  |  :---  |
> > >  | Onlly $\phi$ | $74.37 \pm  0.17$ | $67.90 \pm  0.31$ | $42.91 \pm  0.65$ | $60.04 \pm  0.16$ |
> > >   | RecourseNet $g_\phi$ without $f^{\text{CF}}$| $74.47 \pm  0.19$ | $59.96 \pm  0.00$ | $42.76 \pm  0.12$ | $60.73 \pm  0.09$ |
> > >    |  RecourseNet $g_\phi$ | $76.27 \pm  0.39$ | $68.17 \pm  0.38$ | $45.25 \pm  0.79$ | $62.69 \pm  0.28$ |
> > >
> > > **$100$% Recourse:**
> > >
> > >  |  Method     |  Shapenet-Large  |  Shapenet-Small      |  Speech-Commands  |  Skin-Lesion  |
> > >  |  :---         |  :----    |           :---  |  :---  |  :---  |
> > >  | Onlly $\phi$ | $72.05 \pm  0.17$ | $69.89 \pm  0.65$ | $59.55 \pm  1.55$ | $61.41 \pm  0.26$ |
> > >   | RecourseNet $g_\phi$ without $f^{\text{CF}}$   | $74.43 \pm  0.16$ | $60.12 \pm  0.01$ | $57.27 \pm  0.53$ | $66.85 \pm  0.18$ |
> > >    |  RecourseNet $g_\phi$  | $78.50 \pm  0.26$ | $76.36 \pm  0.58$ | $65.97 \pm  0.90$ | $71.19 \pm  0.71$ |
> > >
> > > We hope that the reviewer will consider these results. Thanks very much!

---

> > > > ### Comment · Reviewer_ERbB · 2022-08-04
> > > > **Thanks for further information**
> > > >
> > > > I appreciate the authors provided further evidence that the performance improvements are statistically significant, although the standard deviations of other baselines are not shown. I hope it will be added in the revision. I will decide my final score after the discussion phase with other reviewers, including the concerns raised by other reviewers.
> > > >
> > > > Thanks!

---

### Author Response · Authors · 2022-08-02
**General Rebuttal comments**

We thank the reviewers for very insightful comments on our work. We have marked all the changes in blue color in our revised Paper and Appendix.

We have reported results on a new medical imaging dataset on skin-lesion detection. In this dataset too, we observe 13.8\% improvement in accuracy of diagnosis with recourse, thus emphasizing the need for recourse. Further, we have incorporated three additional baselines  namely IRM, TARNET, DANN for recourse recommender and two random baselines for Triage. The results and discussion are included in our revised Appendix which further strengthens our proposed approach.

---

### Meta-Review · Area_Chair_2gas · 2022-08-30

**Recommendation:** Accept
**Confidence:** Certain

**Metareview:**

The paper proposes a recourse approach that recommends how to improve performance on instances by modifying their environment. The paper is well motivated and provides a novel approach that is empirically demonstrated to be useful, though the empirical evaluation is limited. Reviewers agree that this paper addresses an important question that has more recently started to get attention, and that the contribution is novel, creative and significant. The quality of the write up could be improved, and I encourage the authors to do so for the camera ready version.

**Award:**

No

---

### Decision · Program_Chairs · 2022-09-14

Accept